# Context effects on probability estimation

**Wei-Hsiang Lin** [1], **Justin L. Gardner** [2,3], **Shih-Wei Wu** [4,5]*

**1** Brain Mind Institute, École Polytechnique Fédérale de Lausanne (EPFL), Lausanne, Switzerland,
**2** Department of Psychology, Stanford University, Stanford, California, United States of America, **3** Wu Tsai Neurosciences Institute, Stanford University, Stanford, California, United States of America, **4** Institute of Neuroscience, National Yang-Ming University, Taipei, Taiwan, **5** Brain Research Center, National Yang-Ming University, Taipei, Taiwan

* swwu@ym.edu.tw

**Data Availability Statement:** Data and analysis code are available in Open Science Framework: https://osf.io/48j7m/.

**Funding:** SWW acknowledges the generous support of Ministry of Science and Technology in Taiwan (most.gov.tw) 101-2628-H-010-001-MY4,

## Abstract

Many decisions rely on how we evaluate potential outcomes and estimate their corresponding probabilities of occurrence. Outcome evaluation is subjective because it requires consulting internal preferences and is sensitive to context. In contrast, probability estimation requires extracting statistics from the environment and therefore imposes unique challenges to the decision maker. Here, we show that probability estimation, like outcome evaluation, is subject to context effects that bias probability estimates away from other events present in the same context. However, unlike valuation, these context effects appeared to be scaled by estimated uncertainty, which is largest at intermediate probabilities. Blood-oxygen-level-dependent (BOLD) imaging showed that patterns of multivoxel activity in the dorsal anterior cingulate cortex (dACC), ventromedial prefrontal cortex (VMPFC), and intraparietal sulcus (IPS) predicted individual differences in context effects on probability estimates. These results establish VMPFC as the neurocomputational substrate shared between valuation and probability estimation and highlight the additional involvement of dACC and IPS that can be uniquely attributed to probability estimation. Because probability estimation is a required component of computational accounts from sensory inference to higher cognition, the context effects found here may affect a wide array of cognitive computations.

## Introduction

When we evaluate a potential reward, such as a job offer, other offers on the table form a unique context that shapes our expectations and influences the way we feel about it. Psychologists and economists model such expectation as a reference point and potential rewards (e.g., different job offers) as gains or losses relative to the reference point [1,2]. Such effects of contexts on valuation impact many decisions we make and are not only observed in humans but also in many other species such as rats [3,4], birds [5], and nonhuman primates [6–8].

However, to make decisions in uncertain environments, organisms not only face the task of evaluating potential rewards but also need to estimate their probabilities of occurrence [9,10]. Understanding how people use probability information has received considerable attention, and people are known to subjectively weight probability information rather than using

104-2410-H-010-002-MY3, 107-2410-H-010-003-MY3, 108-2410-H-010-012-MY3, Ministry of Education in Taiwan: Featured Areas Research Center Program within the Framework of Higher Education Sprout Project (https://sprout.moe.edu.tw/SproutWeb) (grant number: 108BRC-B602) to Brain Research Center at National Yang-Ming University (https://brc.ym.edu.tw/). JLG acknowledges the generous support of Research to Prevent Blindness and Lions Club International Foundation (https://www.rpbusa.org/rpb/low-vision/) and Hellman Fellows Fund (http://www.hellmanfellows.org/). The funders had no role in study design, data collection and analysis, decision to publish, or preparation of the manuscript.

**Competing interests:** The authors have declared that no competing interests exist.

**Abbreviations:** ACC, anterior cingulate cortex; aINS, anterior insula; BIC, Bayesian information criterion; BOLD, Blood oxygen level dependent; dACC, dorsal anterior cingulate cortex; DN, divisive normalization; fMRI, functional magnetic resonance imaging; GLM, general linear model; IPS, intraparietal sulcus; LOSO, leave one subject out; MAG, magnitude of reward outcome; MVPA, multivoxel pattern analysis; OFC, orbitofrontal cortex; RN, range normalization; ROI, region of interest; RPE, reward prediction error; RT, response time; SVR, support vector regression; URD, uncertainty and reference dependent; VMPFC, ventromedial prefrontal cortex; VS, ventral striatum.

objective probability information when making decisions: they tend to overweight small probabilities but underweight moderate to large probabilities. [1,11–16].

But is probability estimation—like valuation—context-dependent? Going back to the job offer example, when one is instead estimating the probability of receiving a potential offer, if there is another offer that is relatively very likely or very unlikely to happen, would it affect how she or he estimates the probability of getting this particular offer? Probability estimate of an event may be affected by other events that take place close in time, which makes it susceptible to context effect in a way similar to how context impacts valuation. However, probability estimation may be different from valuation in that very unlikely or very likely events are less variable than those carrying intermediate probabilities, so that context effect on probability estimation may not be uniform in magnitude across probabilities. Despite such intuitive appeal, standard models of decision making under uncertainty and risk typically do not consider probability estimation to be context-dependent [1,9,10,13], and context effects on probability estimation are seldom investigated.

At the neural implementation level, although previous studies investigated probability and uncertainty coding [17–24], no study had explicitly manipulated context to investigate its effects on probability estimation. Here, we consider whether context-dependent probability estimation would be computed by shared or disassociated neural system for value estimation. The first hypothesis of shared neural implementation centers on the similarity in computations between valuation and probability estimation. That is, if there are context effects on probability estimation, then the similarity with respect to valuation might suggest that both are implemented by the same neural system. In this case, the orbitofrontal cortex (OFC), ventromedial prefrontal cortex (VMPFC), and ventral striatum are candidate regions for context-dependent probability estimation, because accumulating evidence indicate their involvement in subjective-value computations [25–27], in particular the relative and context-dependent representations of subjective value found in these regions [6,28–32]. In particular, Palminteri and colleagues [30] investigated context-dependent computations in value learning and found that a relative, reference-dependent reinforcement learning model better described human subjects' choice behavior and activity in the VMPFC and striatum.

By contrast, the second hypothesis of disassociated neural system suggests that regions that represent context effects on probability estimation do not overlap with those involved in context-dependent valuation. This hypothesis is driven by aspects of probability estimation that are unique to valuation, which involve coding uncertainty and extracting summary statistics of information from the environment. In this case, regions specialized for coding uncertainty and extracting summary statistics, such as mean and variance, from reward history should be highly involved. Previous studies indicate that the dorsal anterior cingulate cortex (dACC), anterior insula (aINS), and intraparietal sulcus (IPS) would be the candidate regions because they had been shown to represent uncertainty-related statistics [17,21] and engage in tracking reward history that is crucial in estimating reward probability [22,23] and decision making under uncertainty and ambiguity [33–35].

To investigate how context impacts probability estimation at the behavioral, computational, and neural implementation levels, we designed a simple stimulus-reward association task in which human subjects were asked to estimate probability of reward associated with visual stimuli through experience. Context was manipulated by pairing stimuli carrying different probabilities of reward in different blocks of trials. We found that, similar to valuation, context effects on probability estimation were reference-dependent. However, unlike valuation, they were scaled by the uncertainty of reward outcomes such that when there was larger uncertainty on which potential outcome would occur (e.g., 50/50 of reward or no reward) there was greater context effect than stimuli with smaller uncertainty (10% or 90% reward). Unexpectedly,

BOLD imaging results showed that both valuation and uncertainty-coding regions are involved in context-dependent probability estimation. Together, these findings point to common neurocomputational substrates shared between probability estimation and valuation but highlight the additional involvement of uncertainty-coding regions that are unique in probability estimation—a central and important cognitive function relevant to a wide array of contemporary problems within neuroscience [36–38].

## Results

In the MRI scanner, subjects ($N$ = 34) performed a stimulus-reward association task in which they were asked to estimate reward probability, but no choice was required (Fig 1A). In each trial, subjects were presented with an abstract visual stimulus that carried a unique probability of reward and asked to estimate its reward probability. After probability estimation, feedback on whether subjects won a monetary reward was provided. In a block of trials, subjects would repeatedly face two different stimuli that appeared in random order. Context was defined by the two stimuli the subjects encountered in a block of trials and was manipulated such that stimuli carrying the same probability of reward were experienced in two different contexts that differed in the probability of reward associated with the other stimulus present in the context (Fig 1B). For example, for 50% reward (middle column in Fig 1B), one stimulus was paired with a stimulus carrying 10% reward (context 1, first row in Fig 1B) and the other with a 90% reward (context 3, third row in Fig 1B). With this design, we were able to manipulate context independently of probability of reward (10%, 50%, and 90%). In Fig 1C, we illustrate the three different contexts used in this experiment with example trial ordering. The goal of the experiment was to investigate how context impacts probability estimation by comparing probability estimates of the stimuli carrying the same reward probability but experienced in different contexts. For example, we want to compare $\hat{P}_{50\%|[10\%,50\%]}$ (probability estimates of the stimulus carrying 50% reward in context 1 in which the other stimulus had 10% reward) with $\hat{P}_{50\%|[50\%,90\%]}$ (probability estimates of the stimulus carrying 50% reward in context 3 in which the other stimulus had 90% reward).

### Context effects on probability estimates

We found significant context effect on probability estimates when reward probability was 50% ($p < 0.05$, nonparametric bootstrap test) but not at 10% and 90% ($p > 0.05$, nonparametric bootstrap test). Subjects gave larger estimates when the 50% reward stimulus was experienced with a 10% reward stimulus than with a 90% reward stimulus (Fig 2A, the two bars in the center of the graph: the bar on the left indicates probability estimate of the 50% reward stimulus [S] when the other stimulus [OS] in the context was 10%; the bar on the right indicates 50% reward probability estimate when OS in the context was 90%). Looking at estimates of individual subjects, 27 out of 34 subjects' estimates were larger in the [10%, 50%] context than in the [50%, 90%] context (Fig 2A, note the direction of tilt of the black lines which represent each single subject's mean probability estimates), showing that most subjects gave larger estimates to the 50% reward stimulus when it was paired with a 10% reward stimulus than with a 90% reward stimulus. Importantly, we observed this effect despite the fact that frequency of reward the subjects experienced was the same between different contexts (two bars in the center of Fig 2B). This indicates that, for the 50% reward stimuli, context effect on probability estimates was not driven by their own reward history. To visualize context effect better for each reward probability, we plot the average difference in probability estimates between contexts (Fig 2C).

We also compared the size of context effect between different probabilities and found that the size of context effect on 50% probability estimate was significantly larger than either 10%

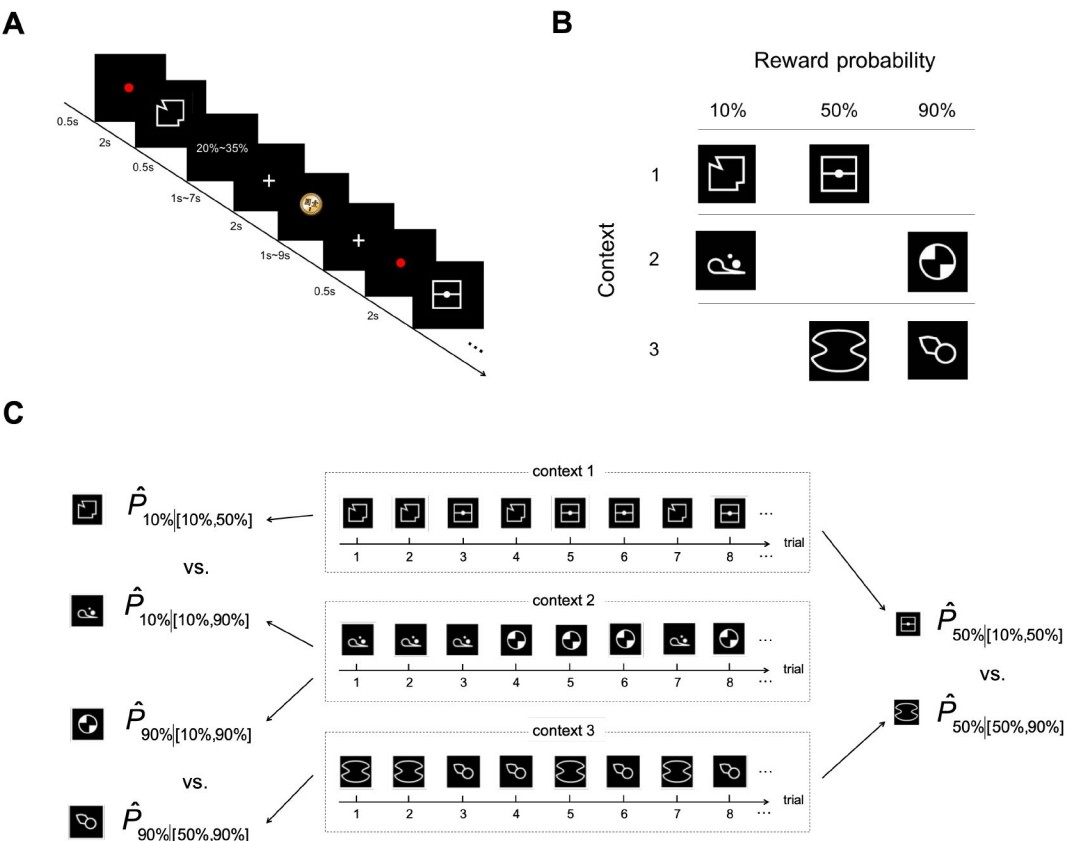

**Fig 1. Task design.** (A) Trial sequence. In each trial, subjects were presented with an abstract visual stimulus and had to indicate his or her interval estimate of its reward probability with a button press. A brief written display (0.5 s) then indicated which probability interval estimate the subject had chosen. After a variable ISI (1–7 s), subjects received feedback on whether she or he received a monetary reward. A variable ITI (1–9 s) was implemented before the start of a new trial. In a block of trials, two stimuli each carrying a unique probability of reward appeared repeatedly in random order. (B) Manipulation of context. Context was defined by the stimuli that appeared in a block of trials. There were three contexts (rows in the table), each consisting of two visual stimuli carrying different probabilities of reward. For each probability, there were two different stimuli assigned to it. The two stimuli with the same probability were experienced under two different contexts. This design allowed us to compare how probability estimates were affected by context. (C) Example trial ordering of the three contexts showing how different stimuli with the same probability of reward could be associated with different contexts. ISI, interstimulus interval; ITI, intertrial interval.

or 90% (Fig 2D). But for 10% and 90%, the difference was not significant. These results were based on trials in the second-half of the functional magnetic resonance imaging (fMRI) session (Fig 2C and 2D). However, the results remained the same regardless of whether we used data from the entire session, the first-half of the session, or the second-half of the session. It is important to note that, even though the effect of context was not statistically significant at 10% and 90%, we did observe a trend that was consistent with the effect on 50%: probability estimate of a stimulus was inversely related to the probability of reward associated with the other stimulus present in the context. At 10% reward, subjects tended to give smaller estimates when the other stimulus carried a 90% than 50% reward (18 out of 34 subjects). At 90% reward, subjects tended to give larger estimates when the other stimulus carried a 10% than 50% reward (23 out of 34 subjects). If we analyzed just the second-half of the data, 20 and 18 subjects showed these results for 10% and 90%, respectively.

At the individual-subject level, we also found evidence for the context effect described above: individual subjects' probability estimates of a stimulus (S) tended to be affected by both

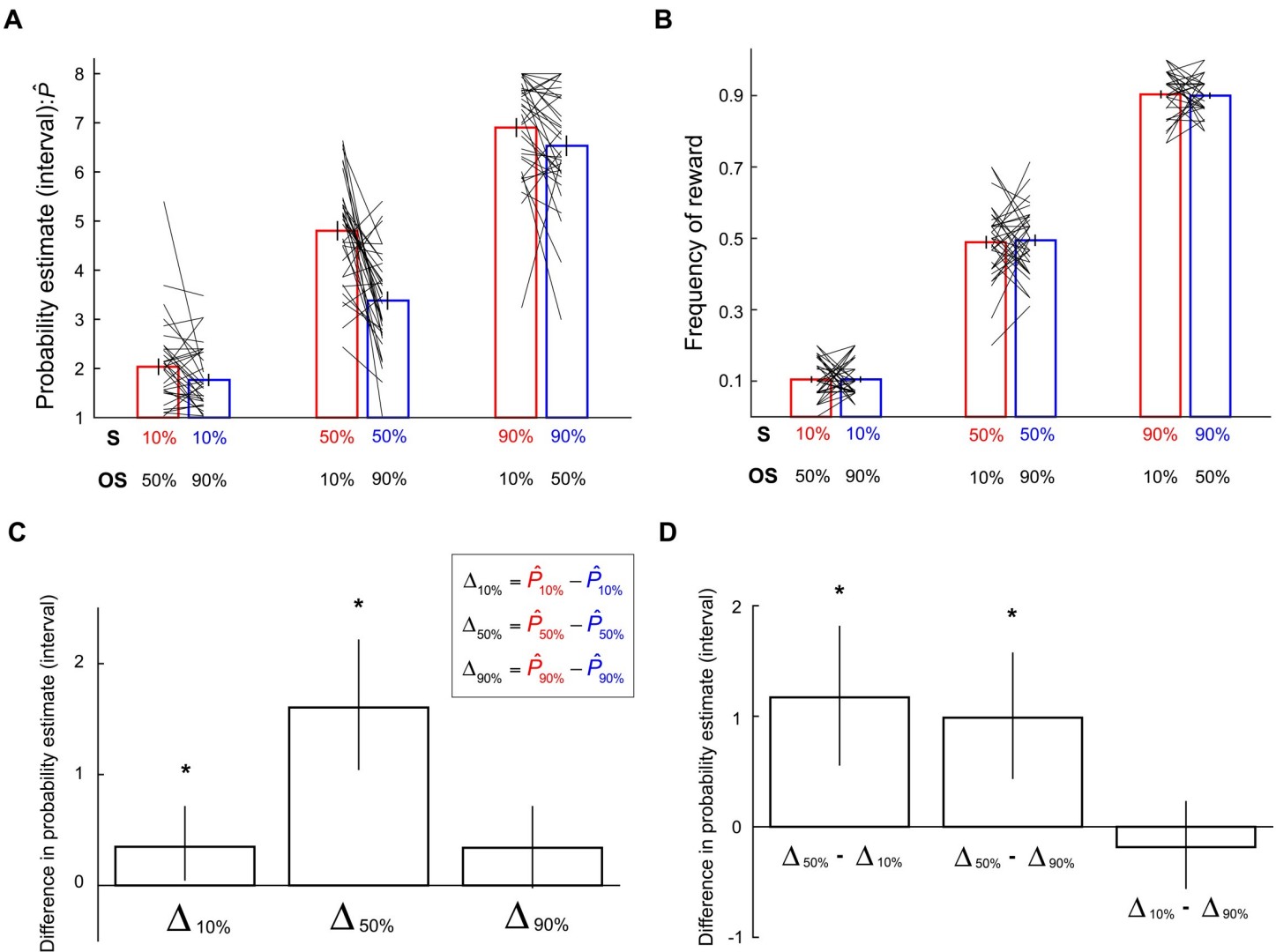

**Fig 2. The effects of context on probability estimates. Experiment 1:** (A) Subjects' estimates of probability of reward associated with different stimuli. Stimuli carrying the same reward probability are plotted together. The vertical axis represents interval estimates: 1:[0%–5%], 2:[5%–20%], 3:[20%–35%], 4:[35%–50%], 5:[50%–65%], 6:[65%–80%], 7:[80%–95%], 8:[95%–100%]. The bars represent the mean probability estimates (across subjects), and the tilted black lines represent individual subjects' data. Error bars represent ±1 SEM. (B) Frequency of reward the subjects experienced during the experiment. Conventions are the same as in Fig 2A. (C) Context-induced difference in probability estimates. For each reward probability—10%, 50%, 90%—the mean difference (across subjects) in probability estimates— $\Delta_{10\%}$, $\Delta_{50\%}$, $\Delta_{90\%}$— between contexts is plotted. In the equations that define $\Delta_{10\%}$, $\Delta_{50\%}$, and $\Delta_{90\%}$ shown in the graph, conventions of the color codes for probability estimates ($\hat{P}$) are the same as in Fig 2A. For example, $\Delta_{50\%}$ is computed by subtracting $\hat{P}_{50\%}$ (in blue), which represents probability estimate of the 50% reward stimulus in the [50%, 90%] context, from $\hat{P}_{50\%}$ (in red), which represents probability estimate of the 50% reward stimulus in the [10%, 50%] context. We obtained $\Delta$ from each subject based on his or her mean probability estimates (across trials) and computed the mean of $\Delta$ across subjects. Error bars represent 95% bootstrap confidence interval of the mean $\Delta$ (across subjects). (D) Comparing the size of context effect between different reward probabilities. Bars indicate the mean difference in context effect (across subjects) for stimuli associated with different probabilities. For example, $\Delta_{50\%}$–$\Delta_{10\%}$ compares context effect between 50% reward and 10% reward. Error bars represent 95% bootstrap confidence interval of the mean differences in context effect (across subjects). Data underlying these graphs can be found in https://osf.io/48j7m. OS, the other stimulus present in the same context as the stimulus of interest; S, stimulus of interest.

the actual reward frequency of S ($f_S$) and the other stimulus ($f_{OS}$) she or he experienced in the same context. For each reward probability, we regressed the mean probability estimates of each subject (across trials) against $f_S$ and $f_{OS}$. For example, for 50% reward, each subject contributed two data points—his or her mean probability estimate of the 50% stimulus in the [10%, 50%] context ($\bar{P}_{50\%|[10\%,50\%]}$) and [50%, 90%] context ($\bar{P}_{50\%|[50\%,90\%]}$). The values of the $f_S$

and $f_{OS}$ regressors for $\bar{P}_{50\%|[10\%,50\%]}$ correspond, respectively, to the reward frequency of the 50% and 10% stimuli; the values of the $f_S$ and $f_{OS}$ regressors for $\bar{P}_{50\%|[50\%,90\%]}$ correspond, respectively, to the reward frequency of the 50% and 90% stimuli. We found that in all three reward probabilities (10%, 50%, 90%), individual subjects' probability estimates were positively correlated with $f_S$ ($p < 0.05$). Consistent with the group average data for 50% reward, individual subjects' probability estimates were negatively correlated with $f_{OS}$ ($p < 0.05$). For 10%, the impact of $f_{OS}$ (negative correlation) was also significant ($p < 0.05$); for 90%, the impact of $f_{OS}$ was not significant ($p = 0.1893$).

## Context-dependent probability estimates predict choice behavior

We also found that context effect on probability estimates further predicted subjects' choice behavior. In a lottery decision task (a behavioral session) that followed the probability estimation task (fMRI session), subjects in each trial faced two stimuli they encountered in the fMRI session and were asked to choose the one they preferred so that one of their choices would be selected at random at the end of the experiment and realized to determine his or her payoff. Each stimulus they faced carried the same reward probability as in the fMRI session, and the reward magnitude was fixed across all stimuli so that subjects should pick the one she or he believed to have the larger probability of reward. Each pair was presented multiple times so that we could calculate choice probability (the fraction of trials in which subjects chose one lottery over the other) for each subject. We analyzed choice probability on the three pairs each consisting of stimuli carrying the same probability of reward but were experienced under different contexts. Subjects' choice behavior was predicted by their context-dependent probability estimates: they preferred the 50% reward stimulus with larger probability estimate (in the [10%, 50%] context) to the one with smaller estimate (in the [50%, 90%] context) (middle bar in Fig 3A): choice probability was significantly larger than 0.5 ($p < 0.05$, nonparametric bootstrap test). In contrast, for the 10% and 90% pairs, subjects were indifferent between the stimuli carrying the same probability of reward (left and right bars in Fig 3A): choice probability was not significantly different from 0.5 ($p > 0.05$, nonparametric bootstrap test). However, it is worth noting that the direction of choice in those pairs was still qualitatively consistent with the probability estimates (Fig 2A): subjects tended to choose more often the stimuli with larger probability estimates.

The impact of context-dependent probability estimates on choice probability was not only observed at the group level—we found that individual differences in probability estimates also predicted choice probability (Fig 3B, 3C and 3D). For the 50% reward pairs, the Pearson correlation between individual subjects' choice probability and probability estimate was 0.4113 ($p = 0.0174$). For 10% and 90% pairs, the correlation was also significant (10%: r = 0.5113, $p = 0.0024$; 90%: r = 0.4629, $p = 0.0067$). This indicates that, for 10% reward and 90% reward, even though the effect of context on probability estimate and on choice probability were not significant at the group level, individual subjects' probability estimate and choice probability are significantly correlated. In summary, these results demonstrate that the effect of context was not only confined to subjects' probability estimates—it further impacted their preferences for the stimuli.

## Dynamics of probability estimates

We examined trial-by-trial probability estimates and found that context effect on 50% reward emerged relatively early and persisted throughout the experiment (Fig 4B). Another aspect of the dynamics worth mentioning is that subjects' probability estimates clearly deviated from the actual frequency of reward that they experienced (the red and blue step-function traces in

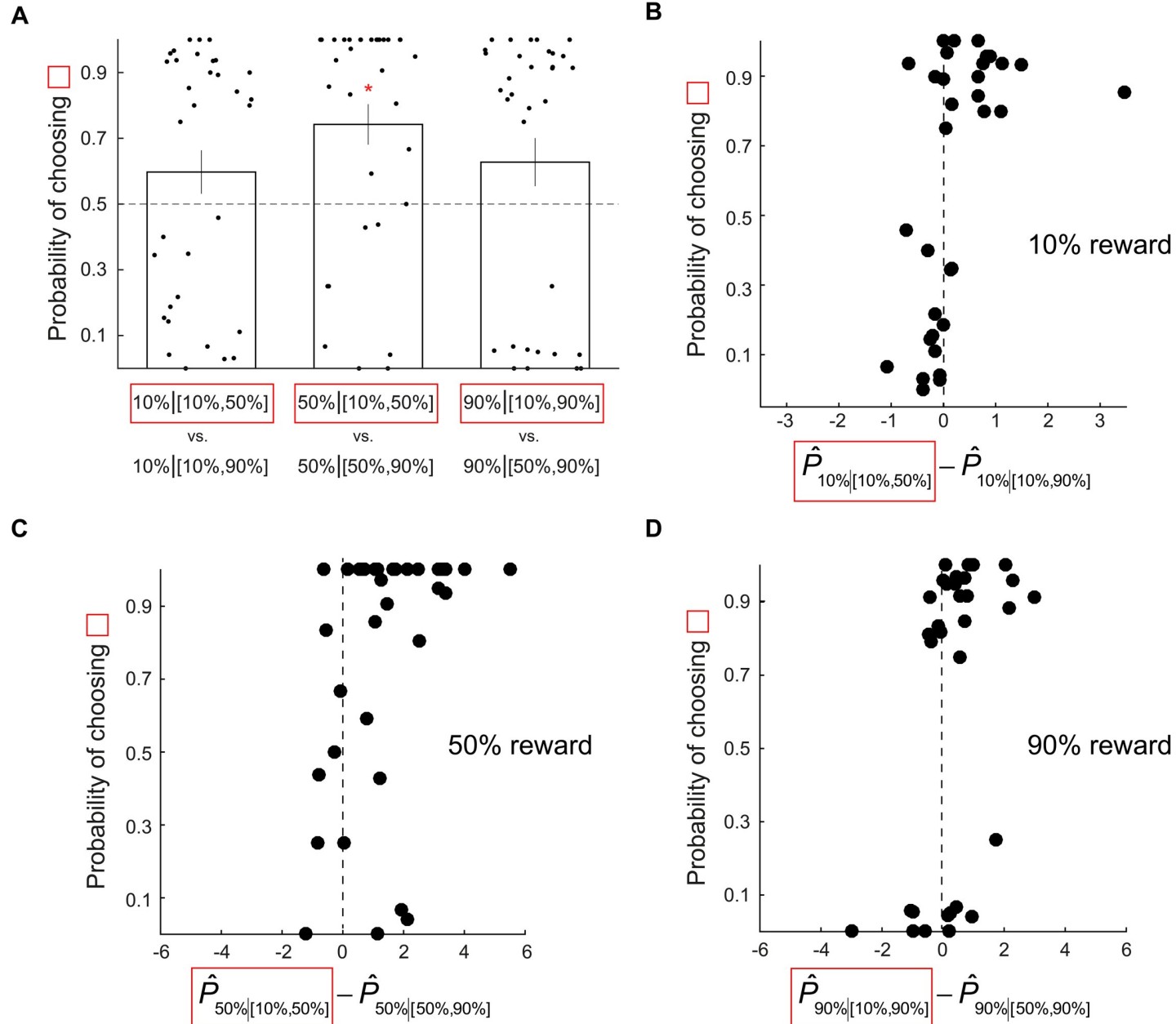

**Fig 3. Experiment 1: Choice probability in the post-fMRI lottery decision task was predicted by context-dependent probability estimates in the fMRI session.** (A) Three pairs of options, each representing a choice between two stimuli carrying the same reward probability but experienced in different contexts, are shown. For each pair, we plot the choice probability of the option highlighted in red. Each data point represents choice probability of a single subject. The bar indicates the mean choice probability averaged across all subjects. Error bars represent ±1 SEM. For each pair, we tested whether the mean choice probability is different from 0.5. The red star symbol indicates significant difference in choice probability from 0.5 based on 95% bootstrap confidence interval. (B–D) Relation between choice probability and probability estimate. For each pair of options described above, choice probability is plotted against the difference in probability estimates subjects provided in the fMRI task. Each data point represents a single subject. (B) 10% reward pair. (C) 50% reward pair. (D) 90% reward pair. Data underlying these graphs can be found in https://osf.io/48j7m.

Fig 4B). Compared with frequency of reward, the subjects clearly underestimated the 50% probability of reward (blue curve) when it was experienced with a 90% reward stimulus in the [50%, 90%] context but overestimated 50% when it was experienced with a 10% reward stimulus in the [10%, 50%] context. In addition, the context effect observed here was not due to

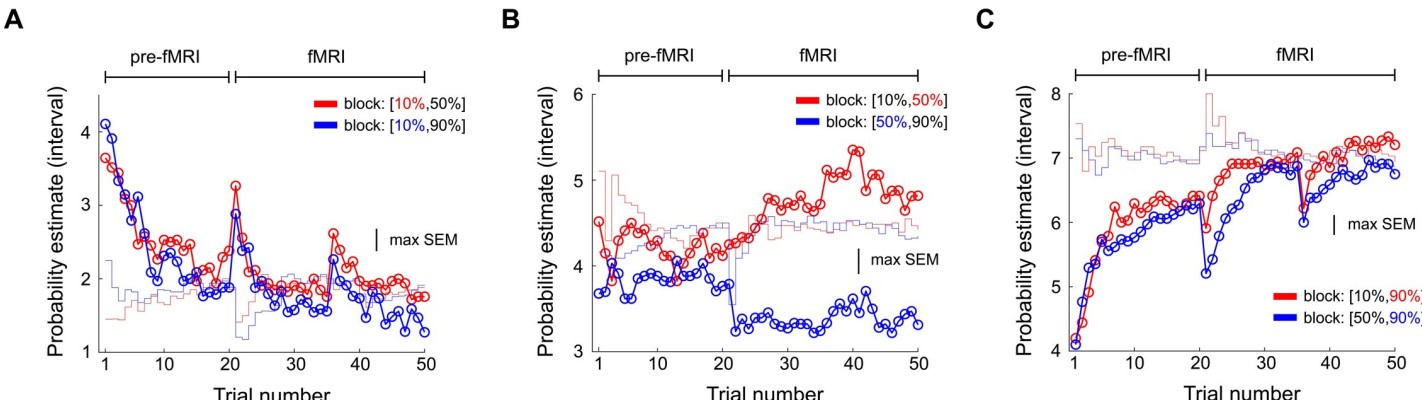

**Fig 4. Experiment 1: Comparison of trial-by-trial probability estimates between contexts showing how differences in probability estimates between contexts developed during the experiment.** Data includes the pre-fMRI practice session and the fMRI session. (A–C) The scale on the vertical axis indicates interval estimate (from 1 to 8 with 1 representing the smallest probability interval [0%–5%] and 8 representing the largest probability interval [95%–100%]). (A) Comparison of the 10% reward stimuli between contexts. Each data point represents the mean probability estimate of a single trial across subjects. Red: 10% stimulus in the [10%, 50%] context. Blue: 10% stimulus is in the [10%, 90%] context. The colored step function traces represent the corresponding mean frequency of reward the subjects experienced. (B) Comparison of the 50% reward stimuli between contexts. Red: 50% stimulus in the [10%, 50%] context. Blue: 50% stimulus in the [50%, 90%] context. (C) Comparison of the 90% reward stimuli between contexts. Red: 90% stimulus in the [10%, 90%] context. Blue: 90% stimulus in the [50%, 90%] context. The bumps on trial number 21 and 36, seen in Fig 4A and 4C, reflect the beginning of a new block of trials. Data underlying these graphs can be found in https://osf.io/48j7m. fMRI, functional magnetic resonance imaging.

reward-frequency bias the subjects experienced in the pre-fMRI session (S2 Fig). By contrast, we did not observe these patterns on the 10% reward (Fig 4A) and 90% reward stimuli (Fig 4C). Qualitatively, however, we did see that subjects gave larger estimates to the 10% reward stimulus when it was experienced with a 50% reward stimulus (red in Fig 4A) than the 10% stimulus experienced with a 90% reward stimulus (blue in Fig 4A). For 90% reward, even though not statistically significant, the subjects gave larger estimates to the 90% reward stimulus in the [10%, 90%] context (red in Fig 4C) than in the [50%, 90%] context (blue in Fig 4C). For response time data, see S1 Fig.

## A behavioral control experiment replicated context effects on probability estimates

We conducted an additional behavioral control experiment (Experiment 2; $N = 20$ subjects) to address a potential confound on the assignment of buttons to indicate probability estimates. In Experiment 1, because the boundary between the two buttons for small probabilities was set at 5% and at 95% for large probabilities instead of 10% and 90%, respectively (see the "Task" subsection in "Materials and methods"), it is possible that the nonsignificant context effect at 10% and 90% was because of a lack of sensitivity on the interval estimates to detect differences in probability estimates at 10% and 90% between different contexts. In the control experiment, a total of 10 buttons were used, each representing a 10% interval and therefore addressed the above concern (see subsection "Behavioral control experiment (Experiment 2)" in "Materials and methods"). We replicated the results observed in Experiment 1: For the 50% reward stimuli, subjects gave larger estimates when 50% was paired with 10% than with 90% (Fig 5B). For the 10% and 90% stimuli, no significant context effect was found (Fig 5A and 5C). Furthermore, as in Experiment 1, context effect and lack thereof on probability estimates predicted subjects' choice behavior in the lottery decision task that followed the probability estimation task (Fig 5D). The subjects on average chose more often the stimulus with larger probability estimate when its reward probability was 50% ($p < 0.05$, nonparametric bootstrap test). By contrast, when stimulus reward probability was 10% and 90%, mean choice probability (across

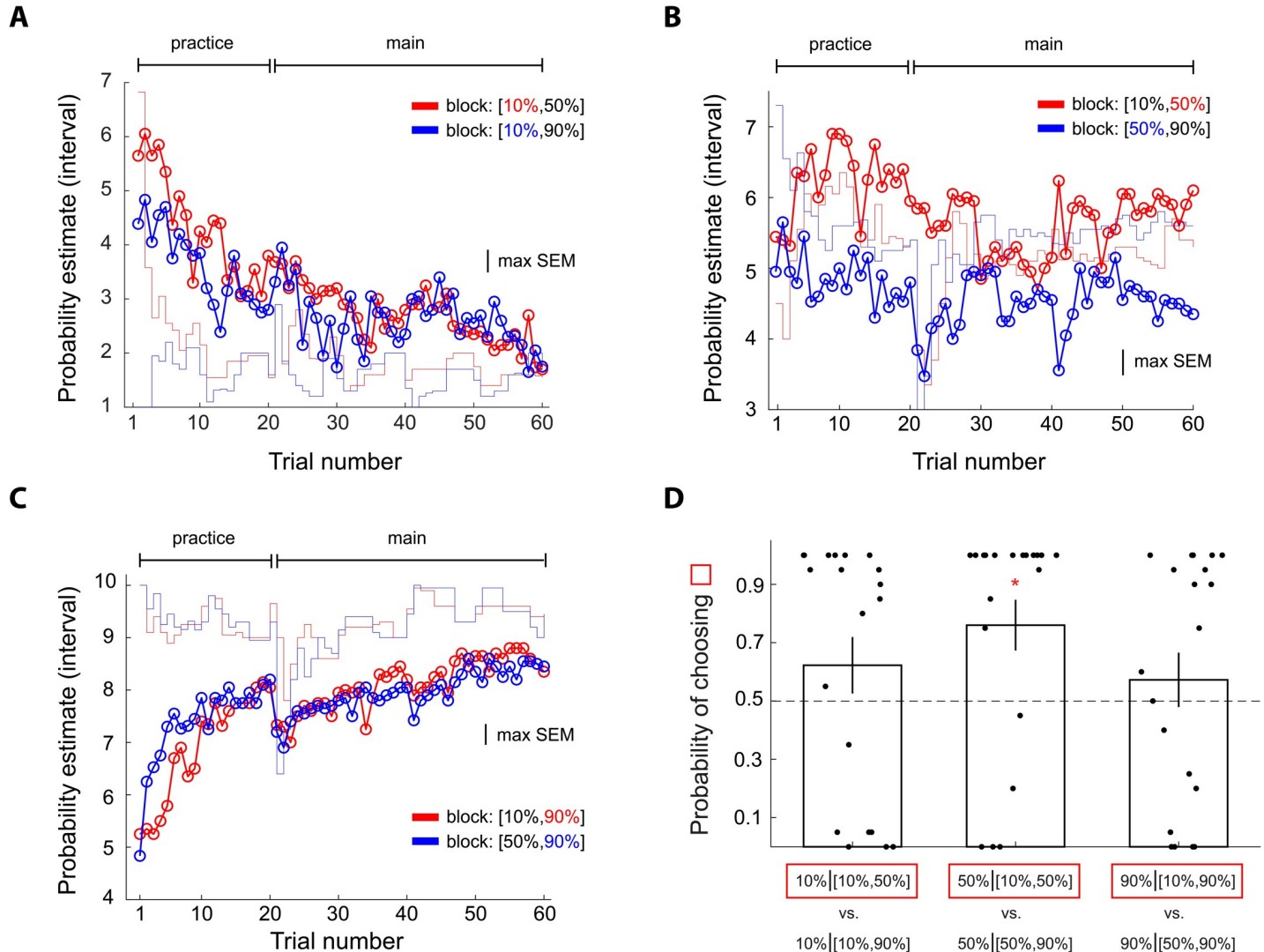

**Fig 5. A behavioral control experiment (Experiment 2) examining whether context effect and lack thereof on probability estimates found in Experiment 1 was because of the design of the interval-to-button mapping.** Conventions are the same as in Figs 3 and 4. The design of the experiment was identical to Experiment 1 except that there were 10—as opposed to 8—key buttons, each representing a unique interval of probability [0, 1] for the subjects to indicate probability estimates. (A) Probability estimates of the 10% reward stimuli. (B) Probability estimates of the 50% reward stimuli. (C) Probability estimates of the 90% reward stimuli. The scale on the vertical axis indicates interval estimate (from 1 to 10 with 1 representing the smallest probability interval [0%–10%] and 10 representing the largest probability interval [90%–100%]). (D) Choice probability in the lottery decision task following the probability estimation task. Error bars represent ±1 SEM. The red star symbol indicates significant difference in choice probability from 0.5 based on 95% bootstrap confidence interval. Data underlying these graphs can be found in https://osf.io/48j7m.

subjects) was not significantly different from 0.5 ($p > 0.05$, nonparametric bootstrap test), indicating that the subjects were indifferent between stimuli carrying the same reward probability but experienced in different contexts.

## Context effects on probability estimates are driven by reference and uncertainty dependencies

To systematically explain context effects, we developed and tested a mathematical model for context-dependent probability estimation (see subsection "Modeling context-dependent computations for probability estimation: the Uncertainty- and Reference-Dependent (URD)

model" in "Materials and methods" and Fig 6A for an illustration). Briefly, the model proposes that when computing probability estimate of a stimulus ($\hat{P}_S$), the decision maker uses the frequency of reward ($f_S$) she or he experienced but is biased by the overall frequency of reward ($f_{overall}$) associated with a context. Here $f_{overall}$ is the frequency of reward averaged across different stimuli in a context. The bias arises from treating $f_{overall}$ as a reference point and comparing $f_S$ with it ($f_S–f_{overall}$). For example, for a stimulus carrying 50% reward, its $f_S$ would be greater than $f_{overall}$ in the [10%, 50%] context in which the other stimulus carries a 10% reward and hence $f_S–f_{overall}>0$, but would be smaller than $f_{overall}$ in the [50%, 90%] context in which the other stimulus carries a 90% reward and hence $f_S–f_{overall}<0$. The model therefore predicts that subjects would give larger probability estimates to the stimulus carrying 50% reward in the [10%, 50%] context than in the [50%, 90%] context because of the reference-dependent difference term ($f_S–f_{overall}$). However, simply having the reference-dependent term alone cannot explain why we did not find significant context effect at 10% and 90% reward. We reasoned that this is because of the level of uncertainty regarding which potential outcome (receiving a reward or no reward) would occur: When the level of uncertainty is larger (e.g., at 50% reward), the impact of ($f_S–f_{overall}$) on probability estimates is greater than when the level of uncertainty is smaller (e.g., at 10% or 90% reward). This is modeled by having ($f_S–f_{overall}$) weighted by the estimated level of uncertainty, which can be either the estimated variance ($\hat{\sigma}_S^2$) or standard deviation ($\hat{\sigma}_S$) of potential outcomes associated with the stimulus based on what the decision maker experienced in the recent past. Below is a version of the model that uses ($\hat{\sigma}_S$) to weight the impact of ($f_S–f_{overall}$)

$$\hat{P}_S = f_S + \hat{\sigma}_S(f_S - f_{overall}). \tag{1}$$

Eq (1) can be rewritten as $\hat{P}_S = (1 + \hat{\sigma}_S)f_S - \hat{\sigma}_S f_{overall}$, which makes it easier to see several predictions of the model on the weights assigned to $f_S$ and $f_{overall}$ (see subsection "Modeling context-dependent computations for probability estimation: the Uncertainty- and Reference-Dependent (URD) model" in "Materials and methods"). These predictions were confirmed by multiple regression analysis on individual subjects' probability estimates using $f_S$ and $f_{overall}$ as regressors (Fig 6B and 6C). First, the regression analysis confirmed that the regression coefficient of $f_S$ was positive and significantly different from 0 for all three reward probabilities (Fig 6B). Second, the regression analysis confirmed that the coefficient of $f_S$ associated with 50% reward was greater than that associated with 10% and 90% reward (Fig 6B). Third, the regression analysis confirmed that the coefficient of $f_{overall}$ associated with 50% reward was negative and its magnitude was larger than that associated with the 10% reward and 90% reward stimuli (Fig 6B). Finally, the model makes the strong prediction that the regression coefficient of $f_S$ ($\hat{\beta}_{f_S}$) would be the same as $1 - \hat{\beta}_{f_{overall}}$, where $\hat{\beta}_{f_{overall}}$ is the regression coefficient of $f_{overall}$. This is what we found (Fig 6C). We tested $\Delta$ against 0, where $\Delta = \hat{\beta}_{f_S} - (1 - \hat{\beta}_{f_{overall}})$ and found that $\Delta$ was not significantly different from 0 in all three reward probabilities ($p > 0.05$, nonparametric bootstrap test). Together, these results provide support to the URD model and highlight that context effects on probability estimates are driven by the interaction of reference-dependent computation and estimated uncertainty associated with potential outcomes.

## Model comparison

We further compared the URD model with normalization models that have been useful in describing context-dependent valuations [32,39,40]. Here, we consider two classes of normalization models, the divisive normalization (DN) model and the range normalization (RN)

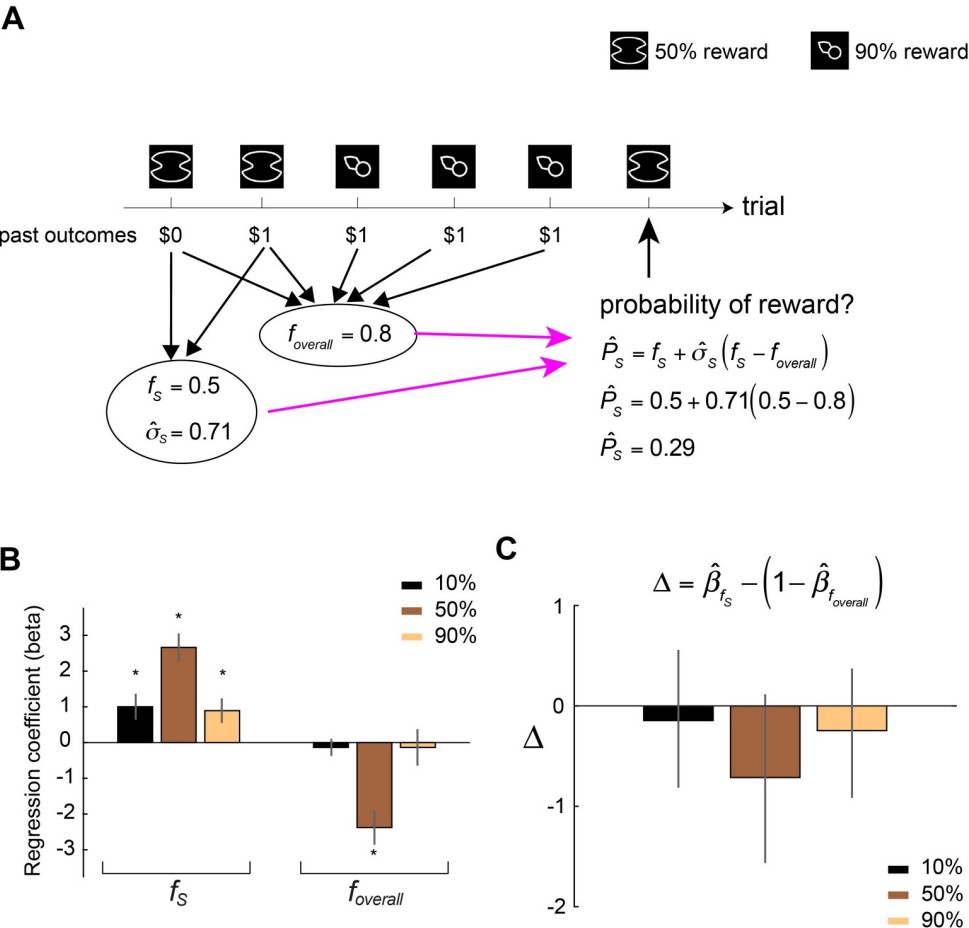

**Fig 6. Testing the URD model for context-dependent probability estimation.** (A) Illustration of the URD model. (B) Regression analysis. In three separate multiple regression analyses each examining a particular reward probability, we estimated the weight subjects assigned to frequency of reward associated with a stimulus ($f_S$) and the overall frequency of reward associated with the context ($f_{overall}$) and plot the mean regression coefficients across all subjects. Error bars represent ±1 SEM. The star symbol indicates statistical significance (testing the mean regression coefficient against 0 at $p < 0.05$). (C) We compare $\hat{\beta}_{f_S}$ with $(1 - \hat{\beta}_{f_{overall}})$ for each reward probability separately. The vertical axis represents $\Delta$, where $\Delta = \hat{\beta}_{f_S} - (1 - \hat{\beta}_{f_{overall}})$. We computed $\Delta$ for each subject separately and plot the mean $\Delta$ across subjects. Error bars represent 95% bootstrap confidence interval of the mean $\Delta$. Data underlying these graphs can be found in https://osf.io/48j7m. URD, uncertainty and reference dependent.

model. DN computes probability estimate of a stimulus by having its frequency of reward divided by the sum of the frequency of reward associated with the two stimuli in a context or divided by the average reward frequency of a context (Eqs 9 and 10 in "Materials and methods"). RN uses the difference between the maximum and the minimum stimulus reward frequency in a context in the denominator to compute probability estimates (Eq 11 in "Materials and methods"). Both model classes predict large context effect on probability estimates at 50% reward and nonsignificant effect at 10%, which are consistent with what we observed in this data set. However, both models also predict large context effect at 90% reward, which we did not observe in subjects' probability estimates. For example, in the simplest form of divisive normalization without free parameters [32]—$\hat{P}_S = f_S/(f_S + f_{OS})$—the normalized value of 10% reward in the [10%, 50%] context is $0.1 \div (0.1 + 0.5) = 0.167$ (suppose that $f_s = 0.1, f_{OS} = 0.5$) and in the [10%, 90%] context is $0.1 \div (0.1 + 0.9) = 0.1$ (suppose that $f_s = 0.1, f_{OS} = 0.9$). The

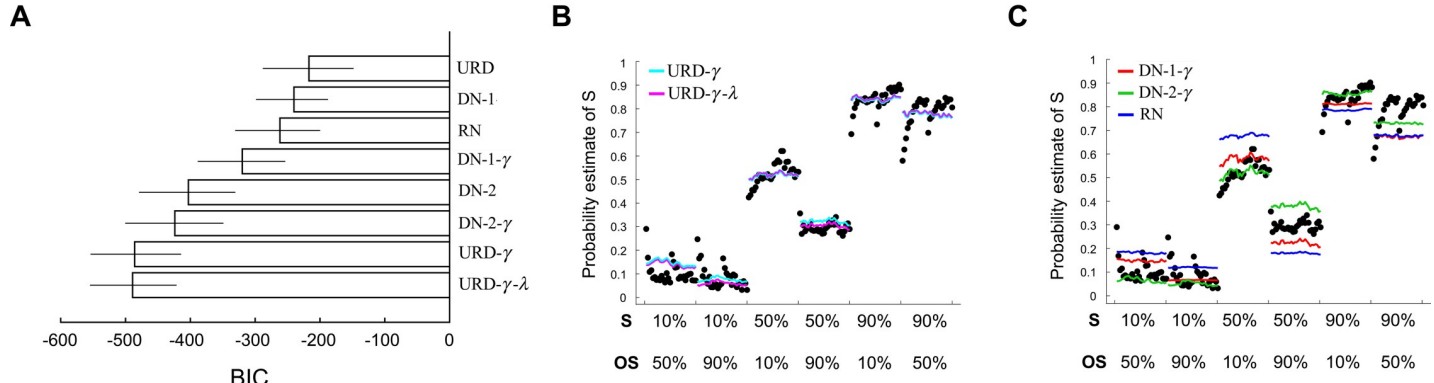

**Fig 7. Model fitting results: Group average data.** (A) Model comparison. We show BIC value of 8 models. Smaller BIC values indicate better models. Error bars represent 95% bootstrap confidence interval. (B) Model fitting results of the URD models (URD-$\gamma$,URD-$\gamma$-$\lambda$). (C) Model fitting results of the divisive normalization models (DN-1-$\gamma$,DN-2-$\gamma$) and range normalization model (RN). Each data point (black) in panels B and C represents the mean probability estimate (across all subjects) of a particular trial order. For example, in the [10%, 50%] context, when S is the 50% reward stimulus, OS would be the 10% reward stimulus; when S is the 10% reward stimulus, OS would be the 50% reward stimulus. Data underlying these graphs can be found in https://osf.io/48j7m. BIC, Bayesian information criterion; DN, divisive normalization; OS, the other stimulus present in the same context as the stimulus of interest; RN, range normalization; S, stimulus of interest; URD, uncertainty and reference dependent.

difference is 0.067, a small difference. By contrast, for 90% reward, under [10%, 90%] the normalized value is $0.9 \div (0.9 + 0.1) = 0.9$ (suppose that $f_s = 0.9, f_{OS} = 0.1$); under [50%, 90%] the normalized value is $0.9 \div (0.9 + 0.5) = 0.643$ (suppose that $f_s = 0.9, f_{OS} = 0.5$). This indicates that DN would in principle predict large context effect at 90% reward but small effect at 10% reward. It is also worth mentioning that, in contrast to the URD model, both DN and RN models do not use information about reward variance or standard deviation when computing probability estimates.

A total of 9 different models—three versions of our model (URD), four versions of DN, and two versions of RN—were fitted to the trial-by-trial mean probability estimates (across subjects) using method of maximum likelihood (see subsection "Model fitting and model comparison" in "Materials and methods"). To compare these models, we computed Bayesian information criterion (BIC) and for each model estimated its 95% confidence interval using nonparametric bootstrap method [41] (Fig 7A). We did not include the BIC of RN-$\gamma$ because of poor convergence in maximum likelihood estimation. The best models are the versions that considered probability distortion—overestimation of small probability and underestimation of large probability (see Eq 6 in "Materials and methods"; model labels including $\gamma$ in Fig 7A). Among them, there was no significant difference between our model (URD-$\gamma$,URD-$\gamma$-$\lambda$) and DN models (DN-1-$\gamma$,DN-2-$\gamma$,DN-2). However, qualitatively, compared with our model fits (Fig 7B), the normalization models (Fig 7C) tended not to capture the data as well across different probabilities and contexts, especially at 90% reward.

We also fitted the same models at the individual-subject level (Fig 8A and 8B). The overall pattern of the model fits was very similar to the results from fitting the group average data. Finally, we implemented all the models considered above in the Rescorla–Wagner reinforcement-learning model framework and fitted them (S1 Text). We fit these models at the individual-subject level and plot the group average model fits (Fig 8C and 8D). The Rescorla–Wagner version of these models captured the dynamics of average probability estimates better compared with their non-Rescorla–Wagner counterparts (Fig 8A and 8B). Further, under the Rescorla–Wagner model framework, it is no longer obvious that our model described the data better than normalization models at 50% and 90% reward. However, these differences could

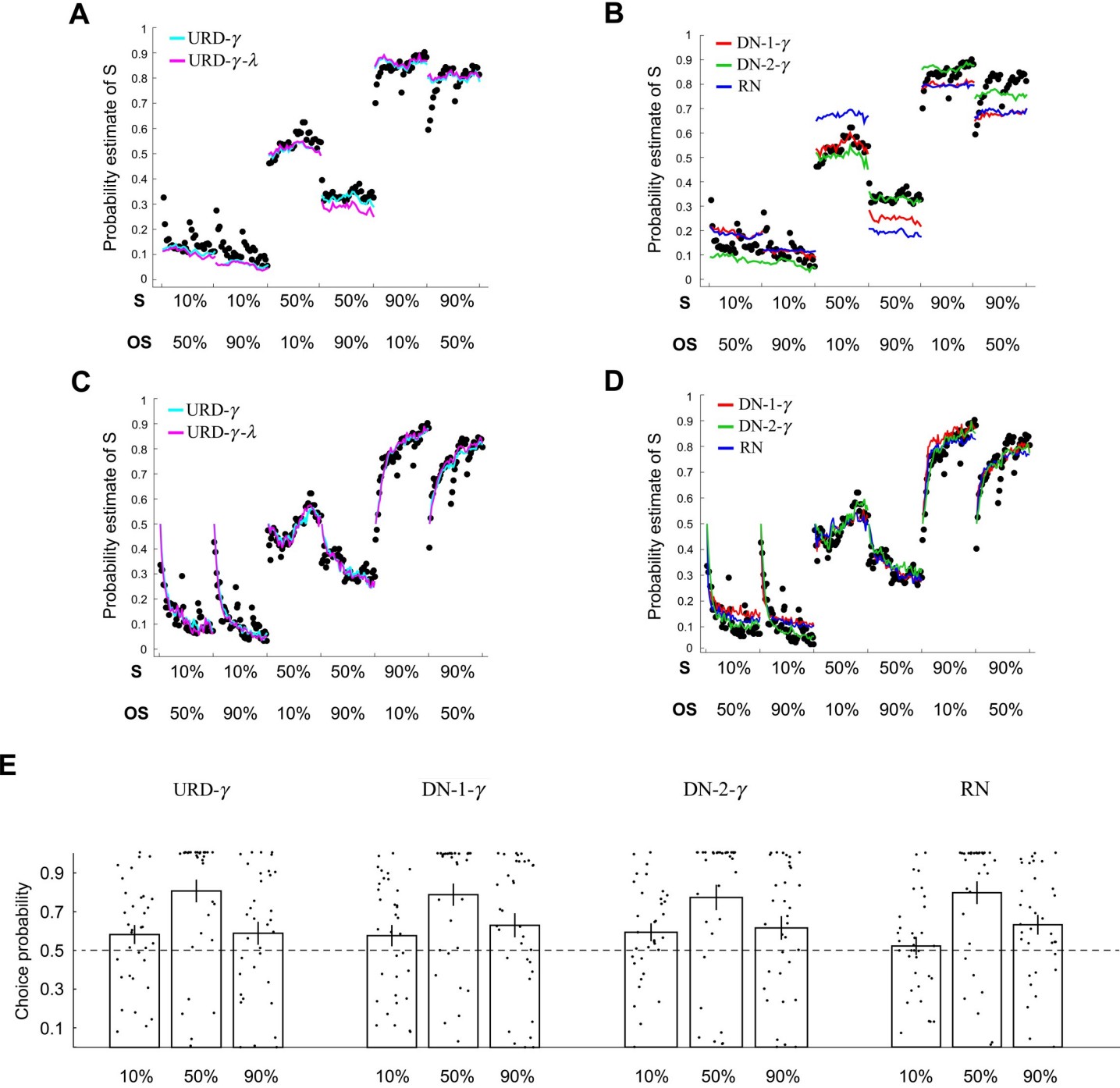

**Fig 8. Model fitting results: Individual data.** (A–B) Results from fitting the models at the individual-subject level. The mean model fits (averaged across subjects) are plotted along with subjects' average probability estimate. (A) Results from URD models (URD-$\gamma$,URD-$\gamma$-$\lambda$). (B) Results from DN models (DN-1-$\gamma$,DN-2-$\gamma$) and RN model (RN). (C–D) Results from fitting the models implemented in the Rescorla–Wagner reinforcement-learning model framework. Models were fit at the individual-subject level. (C) Results from URD models (URD-$\gamma$,URD-$\gamma$-$\lambda$). (D) Results from DN models (DN-1-$\gamma$,DN-2-$\gamma$) and RN model (RN). (E) Model simulations on choice probability. Here, we show simulations from four of the models mentioned above. We plot choice probability based on simulated choice data according to the Rescorla–Wagner model parameter estimates from each subject. These results should be compared with subjects' actual choice probability shown in Fig 3A. Conventions are the same as in Fig 3A. Data underlying these graphs can be found in https://osf.io/48j7m. DN, divisive normalization; OS, the other stimulus present in the same context as the stimulus of interest; RN, range normalization; S, stimulus of interest; URD, uncertainty and reference dependent.

be because of an increase in the number of free parameters in the Rescorla–Wagner versions of these models as a result of having to incorporate learning-rate parameters and our choice to fit each context separately rather than altogether (a strategy adopted in fitting the non-Rescorla–Wagner version of the models). Nonetheless, our model qualitatively described probability estimates equally well, if not better, compared with normalization models when fitting the group data, when fitting individual-subject data, and when fitting individual-subject data using Rescorla–Wagner version of the models. We also fitted other versions of these three model classes, namely, our URD model, the DN model, and the RN model. They in general did not produce significantly different results (S4 Fig).

To show that the model fits can capture subjects' choice behavior in the lottery decision task (post-fMRI session; Fig 3A), we simulated each subject's choice behavior based on his or her model parameter estimates in the Rescorla–Wagner model framework and computed choice probability (Fig 8E). In general, all the four models illustrated here captured subjects' actual choice probability equally well.

In summary, although the three model classes considered here—the URD model, DN model, and RN model—did not significantly differ in model comparison statistic, qualitatively the URD model tended to capture the context effect on probability estimates and lack thereof across different reward probabilities better.

## fMRI results

We performed both univariate general linear model (GLM) analysis and multivoxel pattern analysis (MVPA) on the fMRI data. All the analyses focused on two time windows during a trial—when the visual stimulus was presented and subjects had to estimate reward probability (stimulus presentation) and when the reward outcome (whether subjects received a reward or no reward) was shown (reward feedback). For whole-brain GLM analysis, we performed cluster-level inference using Gaussian random field theory (familywise error corrected at $p < 0.05$) with a $p < 0.001$ ($z > 3.1$) cluster-forming threshold.

## Average brain activity did not represent context effect on probability estimates

We found no evidence for neural representations of context effect on probability estimates based on average brain activity in response to different stimuli. First, at stimulus presentation, for each reward probability separately, we compared activity between different contexts and got null results at the whole-brain level. Second, we examined whether there were regions representing individual differences in context effect on 50% reward (the probability that showed significant context effect at the behavioral level). For each subject separately, we computed the difference in the mean probability estimates (across trials) between contexts ($\Delta \hat{P}_{50\%}$) and used it as a behavioral measure of context effect. We performed a group-level mixed-effect analysis on the difference in average brain activity in response to 50% reward stimuli between different contexts using individual subjects' $\Delta \hat{P}_{50\%}$ as a covariate and found no significant result at the whole-brain level. Third, region-of-interest (ROI) analysis on the valuation network in the VMPFC and ventral striatum (VS; using coordinates from the work by Clithero and Rangel [27]) also did not show significant result on the two analyses described above. Together, these findings indicate that context effect on probability estimates is not represented by stimulus-specific average brain activity (across trials) at the time of stimulus presentation. For results at the time of reward feedback, see "Replication of reward magnitude and prediction error signal in VMPFC and striatum" below.

## Multivoxel patterns of activity in the dorsal anterior cingulate cortex, ventromedial prefrontal cortex, and intraparietal sulcus predict individual differences in context effect on probability estimates

We subsequently examined whether context effect on probability estimation can be represented in patterns of multivoxel activity by testing whether we can predict individual differences in context effect based on patterns of multivoxel activity. A between-subject, cross-validated Multivoxel Pattern Analysis (MVPA) was performed using a searchlight-based approach [42] looking at predefined ROIs in dACC, VMPFC, VS, aINS, and IPS. Among them, VMPFC, VS, and dACC are involved in representing subjective value in value-based decision making [26,27] and uncertainty-related statistics. The IPS and aINS, although less mentioned in the value-based decision making literature, had also been shown to be involved in representing probability and uncertainty in decision making [35].

We analyzed multivoxel activity patterns separately at the time of stimulus presentation and reward feedback. We found that, at both time windows, multivoxel patterns of dACC activity significantly predicted subjects' context effect at 50% reward (Fig 9A, 9B and 9C). Multivoxel pattens of VMPFC activity predicted context effect at the time of reward feedback (Fig 9D and 9E), and multivoxel patterns of right IPS predicted context effect at the time of stimulus presentation (Fig 9F). For each subject separately, we computed the behavioral measure of context effect—the difference in mean probability estimates ($\Delta \hat{P}_{50\%}$) between contexts: mean probability estimate (across trials) of 50% reward stimulus in the [10%, 50%] context minus the mean probability estimate (across trials) of 50% reward stimulus in the [50%, 90%] context. For each subject separately, we computed the neural measure of context effect—the difference in mean BOLD response to the 50% stimuli between the [10%, 50%] context and [50%, 90%] context, which we estimated for each voxel separately. In the searchlight (a sphere with 8-mm radius) analysis, we trained a support vector regression (SVR) to learn and predict individual subject's $\Delta \hat{P}_{50\%}$ based on multivoxel patterns of activity within the searchlight and performed $n$-fold ($n$ = 33 subjects) cross-validation. The correlation between the left-out subjects' $\Delta \hat{P}_{50\%}$ and the SVR-predicted $\Delta \hat{P}_{50\%}$ was then computed. To assess whether prediction performance was above chance, we performed nonparametric permutation test as in the work by Schmack and colleagues [42]. For each searchlight, we constructed the null distribution of correlation coefficients by repeatedly doing the following: we randomly permuted the labels ($\Delta \hat{P}_{50\%}$), trained the SVR based on them, and computed the correlation between actual and predicted $\Delta \hat{P}_{50\%}$. Prediction accuracy was considered significant if the probability of the true correlation was less than 0.05 Bonferroni-corrected for multiple comparisons given the number of voxels tested in an a priori ROI. For dACC, at stimulus presentation, 8 voxels survived Bonferroni correction (magenta voxels in Fig 9A); at reward feedback, 5 voxels survived Bonferroni correction (cyan voxels in Fig 9A). For VMPFC, 8 voxels survived Bonferroni correction; for right IPS, one voxel survived Bonferroni correction. The scatter plot in Fig 9B shows the result from the peak dACC voxel—searchlight centering on this voxel produces the largest correlation between predicted $\Delta \hat{P}_{50\%}$ and actual $\Delta \hat{P}_{50\%}$—at stimulus presentation (r = 0.89, [x2, y30, z18]). The plots in Fig 9C and 9E, respectively, show the results from the peak dACC voxel at reward feedback (r = 0.83, [x2, y24, z20]) and the peak VMPFC voxel at reward feedback (r = 0.85, [x −12, y46, z0]). The plot in Fig 9F shows result from the peak right IPS at stimulus presentation (r = 0.82, [x48, y−42, z44]). The dashed lines (Fig 9B, 9C, 9E and 9F) represent perfect prediction, not regression fits.

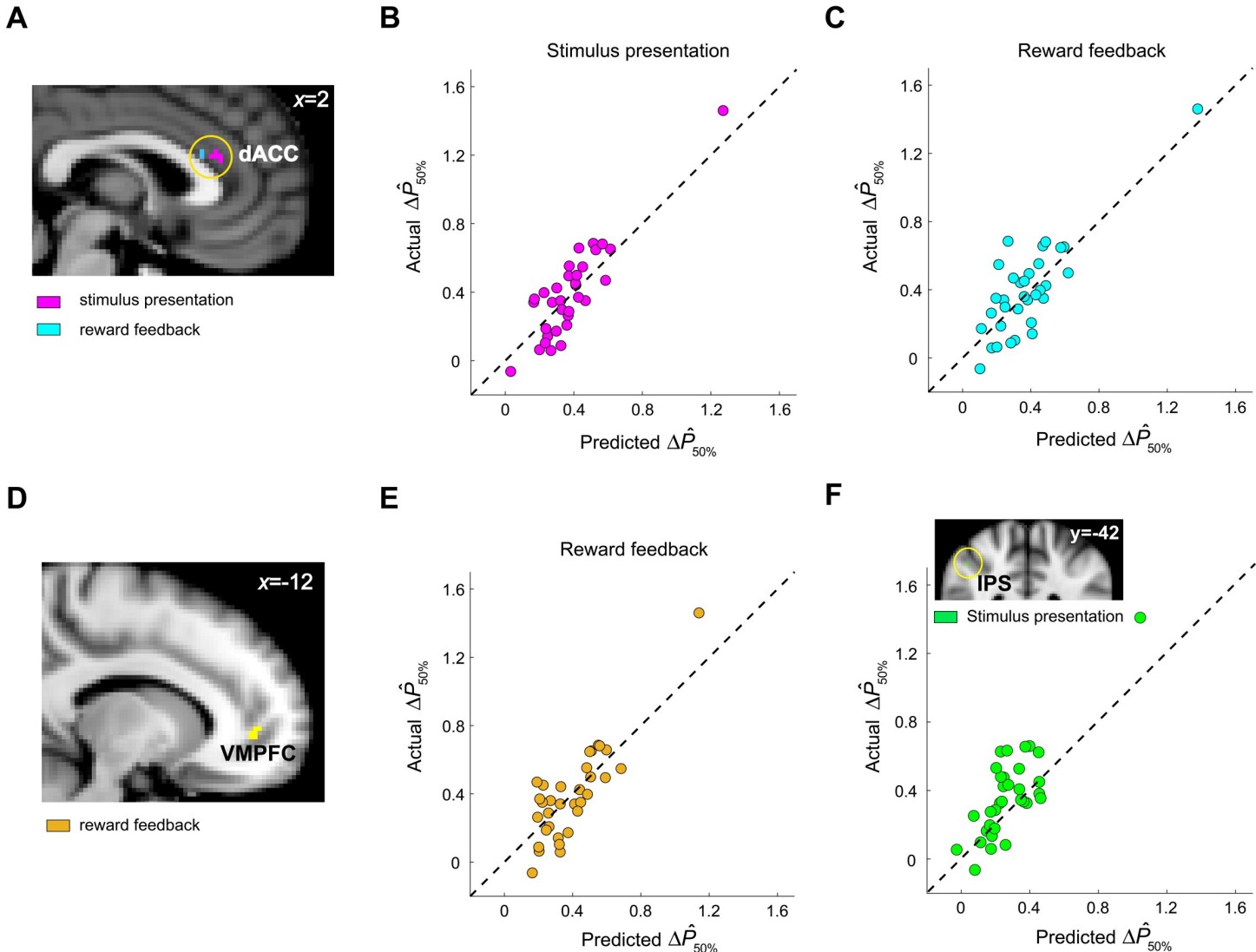

**Fig 9. dACC, VMPFC, and rIPS represent context effect on probability estimates.** (A–C) dACC results. (A) In a between-subject, cross-validated MVPA, we found that patterns of multivoxel activity in dACC predicted context effect on individual subjects' probability estimates ($\Delta \hat{P}_{50\%}$) using activity patterns at the time of stimulus presentation (magenta) and at reward feedback (cyan; $p < 0.05$, Bonferroni corrected for 1,350 voxels in the dACC ROI). The voxels shown are the ones that survived Bonferroni correction. (B) Actual $\Delta \hat{P}_{50\%}$ plotted against predicted $\Delta \hat{P}_{50\%}$ based on activity pattern—at the time of stimulus presentation—in the searchlight centered on the dACC voxel that produced the largest correlation between actual and predicted $\Delta \hat{P}_{50\%}$ (referred to as the peak voxel). (C) Actual $\Delta \hat{P}_{50\%}$ plotted against predicted $\Delta \hat{P}_{50\%}$ based on activity pattern—at the time of reward feedback—of the peak voxel in dACC. (D–E) VMPFC results. (D) VMPFC voxels that significantly predicted behavioral context effect ($\Delta \hat{P}_{50\%}$) at the time of reward feedback ($p < 0.05$, Bonferroni corrected for 1,776 voxels in the VMPFC ROI). The voxels shown are the ones that survived Bonferroni correction. (E) Scatter plot using data from the peak VMPFC voxel. Conventions are the same as in panels B and C. (F) rIPS results ($p < 0.05$, Bonferroni corrected for 196 voxels in the rIPS ROI). Scatter plot using data from the peak rIPS voxel. The dashed line in panels B, C, E, and F represents the 45-degree line, indicating perfect prediction. Data underlying these graphs can be found in https://osf.io/48j7m. dACC, dorsal anterior cingulate cortex; MVPA, multivoxel pattern analysis; rIPS, right intraparietal sulcus; ROI, region of interest; VMPFC, ventromedial prefrontal cortex.

## Replication of reward magnitude and prediction error signal in VMPFC and striatum

Examining brain activity at the time of reward feedback, we replicated previous findings on reward magnitude [26] and reward prediction error (RPE) [18,43–47] representations in the VMPFC and VS (Fig 10). In a closer look at VMPFC representations for reward magnitude

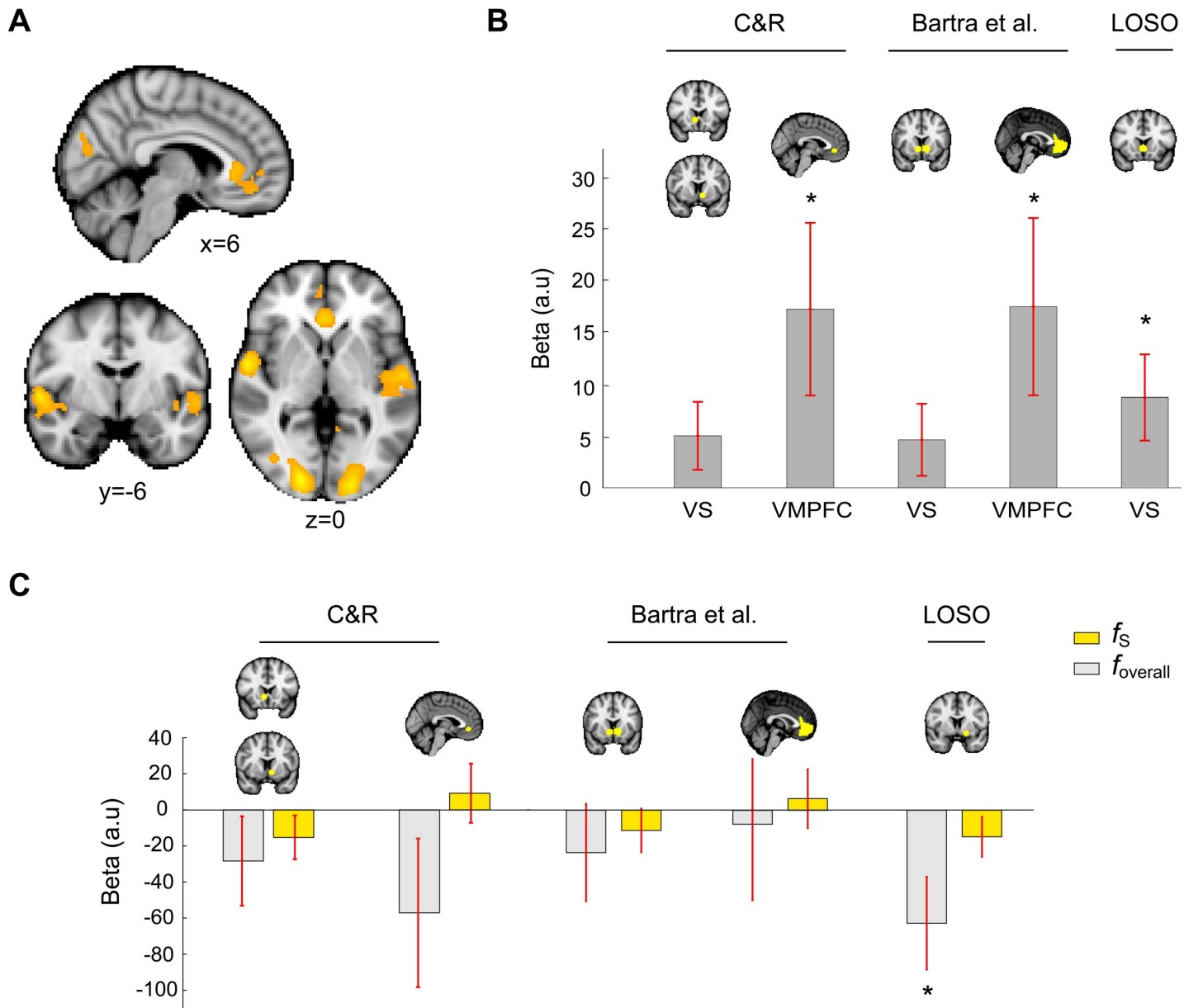

**Fig 10. Univariate GLM results replicating reward magnitude and reward prediction error representations in VMPFC and VS and showing significant reward-frequency representation in VS.** (A) Neural correlates of reward magnitude at the time of reward feedback. Cluster-level inference using Gaussian random field theory was performed (familywise error corrected at $p < 0.05$) with a $p < 0.001$ cluster-forming threshold. (B) ROI analysis on reward prediction error. We used VMPFC and VS ROIs from two meta-analysis papers on value-based decision making ([26]: shown as Bartra and colleagues; [27] shown as C&R) and VS ROI identified through LOSO method based on reward prediction error contrast at the time of reward feedback. (C) ROI analysis on frequency of reward associated with stimulus ($f_S$) and the overall reward frequency associated with a context ($f_{overall}$). *$p < 0.05$. Data underlying these graphs can be found in https://osf.io/48j7m. GLM, general linear model; LOSO, leave one subject out; ROI, region of interest; VMPFC, ventromedial prefrontal cortex; VS, ventral striatum.

with a larger cluster-forming threshold ($z > 3.6$), we found two separate clusters—one in the medial OFC (maximum z statistic = 3.76, [x8, y44, z−12]) and the other in the ACC (maximum z statistic = 4.31, [x2, y34, z2]; Fig 10A). We also found that VMPFC and VS represented RPE—the difference between reward outcome (1 = reward, 0 = no reward) and subjects' trial-by-trial probability estimates (Fig 10B; ROI analysis). We used VMPFC and VS ROIs based on two previous meta-analysis papers on subjective-value representations in decision making (Fig

10B and 10C: Bartra and colleagues [26]; Clithero and Rangel [27]. We also used leave-one-subject-out (LOSO) method to construct ROI in VS based on the RPE contrast in GLM-1 (see subsection "General linear modeling of BOLD response" in "Materials and methods").

## Representation of reward-frequency statistics in ventral striatum

We examined the neural representations for frequency of reward associated with the stimulus ($f_S$) and the overall frequency of reward ($f_{overall}$)—two statistics critical to context-dependent probability estimation in our computational model. At the whole-brain level, we did not find any region that correlated with either one of these statistics. However, in ROI analysis we found that VS activity negatively correlated with $f_{overall}$ (Fig 10C, LOSO ROI). The negative correlation was consistent with what we found in the behavioral analysis showing that $f_{overall}$ was negatively correlated with subjects' probability estimates (Fig 6B).

## Four additional behavioral experiments

To further explore probability estimation at different reward probabilities and contexts, we conducted four additional behavioral experiments. In the previous two experiments (fMRI experiment—Experiment 1; behavioral control experiment—Experiment 2), we investigated context effects on three reward probabilities—10%, 50%, and 90%. In the new experiments, we included 30% and 70% that were not investigated in Experiments 1 and 2. These new experiments also allowed us to create new contexts to look at the original probabilities (10%, 50%, and 90%). We note that in Experiments 3 to 5 the probabilities examined were much closer to each other than Experiments 1 and 2. In these situations, the URD model would predict smaller context effects that might not reach statistical significance. Hence, we discussed these results qualitatively in terms of whether they were consistent with the model prediction.

In Experiment 3, we examined reward probabilities at 10%, 30%, and 50% ($N$ = 20 subjects). The design principle was identical to Experiments 1 and 2 (Fig 1). Hence, three contexts—[10%, 30%], [10%, 50%], and [30%, 50%]—were implemented. In addition to examining context effect on 30% that was not investigated in Experiments 1 and 2, this experiment was further motivated by the following reasons. First, this experiment allowed us to examine 10% in a new context—[10%, 30%]. Second, different from Experiments 1 and 2 in which the 50% reward stimulus was the better stimulus in one context ([10%, 50%])—in terms of how likely subjects would receive a reward—but was the worse stimulus in the other ([50%, 90%] context), in this experiment the 50% reward stimuli always had the best prospect (compared with 10% and 30% reward). This experiment thus allowed us to further examine whether context effect on 50% (Experiments 1 and 2) can be purely driven by a better/worse asymmetry. In other words, if context effect on 50% were purely driven by better/worse asymmetry, we would not expect to see context effect on 50% in this experiment. By contrast, if context effect on 50% were driven by the interaction of reference-dependent computation and estimated uncertainty as predicted by the URD model, we would expect to see an effect of context on 50% reward in this experiment.

The results of Experiment 3 were consistent with our model prediction. For 30% reward, subjects gave larger probability estimates (red in Fig 11B) in the context in which the 30% reward stimulus was the better (more likely to receive a reward) stimulus ([10%, 30%] context) than in the context in which it was the worse stimulus ([30%, 50%] context; blue in Fig 11B). For 50% reward, subjects gave larger probability estimates in the [10%, 50%] context than in the [30%, 50%] context (Fig 11C). Because 50% was always the best stimulus, this pattern cannot be explained purely based on the better/worse asymmetry. By contrast, for 10% reward, probability estimates were virtually identical between different contexts (Fig 11A). Finally, we

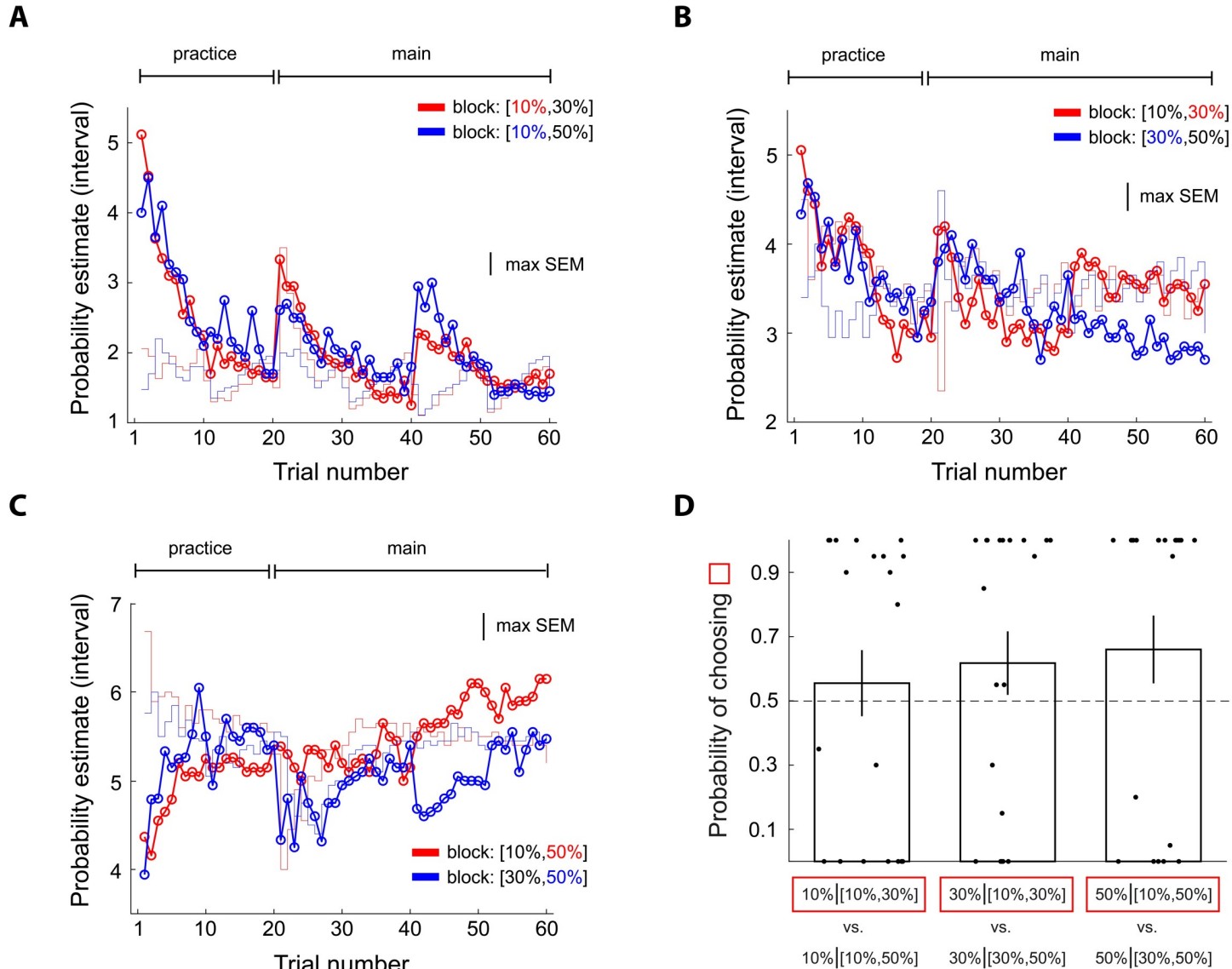

**Fig 11. Experiment 3: Examining context effects on probability estimates at 10%, 30%, and 50% reward and finding further support for the URD model.**
Conventions are the same as in Fig 5. The design principle of the experiment was identical to the design in Fig 1B except that subjects faced 10%, 30%, and 50% reward probabilities instead of 10%, 50%, and 90% reward. (A) Probability estimates of 10% reward. (B) Probability estimates of 30% reward. (C) Probability estimates of 50% reward. (D) Choice probability in the lottery decision task after the probability estimation task. Error bars represent ± 1 SEM. Data underlying these graphs can be found in https://osf.io/48j7m. URD, uncertainty and reference dependent.

found that subjects' choice behavior in the lottery decision task was consistent with their probability estimates (Fig 11D).

In Experiment 4, we examined reward probabilities at 50%, 70%, and 90% ($N$ = 20 subjects). The design principle was identical to Experiments 1 and 2 (Fig 1). Hence, three contexts— [50%, 70%], [50%, 90%], and [70%, 90%]—were implemented. This experiment thus allowed us to investigate context effect on a new reward probability at 70%. In addition, similar to the motivations for Experiment 3, we used this experiment to examine 50% and 90% that were present in Experiments 1 and 2 in greater depth. First, in contrast to Experiments 1 and 2 in which 50% was the better stimulus—in terms of how likely it was for subjects to get a reward—

in one context and the worse stimulus in the other, here, the 50% reward stimuli always had the worst prospect (compared with 70% and 90% reward). Together with Experiment 3, these two experiments provided us the opportunity to further examine whether context effect on 50% observed in Experiments 1 and 2 arose from the interactions of uncertainty and reference-dependent computations or can simply be attributed to the better/worse asymmetry. Second, this experiment also allowed us to investigate 90% in greater depth by examining 90% in a new context—[70%, 90%]—in addition to the [10%, 90%] and [50%, 90%] contexts investigated in Experiments 1 and 2.

The results of Experiment 4 were consistent with our model prediction. For 70% reward, subjects gave larger probability estimates in the [50%, 70%] context in which it was the better stimulus (red in Fig 12B) than in the [70%, 90%] context where it was the worse stimulus (blue in Fig 12B). For 50% reward, subjects gave smaller probability estimate when the other stimulus carried a 90% reward ([50%, 90%] context; blue in Fig 12A) than when the other stimulus carried a 70% reward ([50%, 70%] context; red in Fig 12A). Because the 50% reward stimuli always had the worst prospect, the effect of context observed here cannot be simply attributed to the better/worse asymmetry. By contrast, for 90% reward, probability estimates were virtually identical between different contexts (Fig 12C). Finally, we found that subjects' choice behavior in the lottery decision task was consistent with their probability estimates (Fig 12D).

In Experiment 5, we investigated reward probabilities at 30%, 50%, and 70% reward ($N = 20$ subjects). Similar to Experiments 1 and 2, 50% was the better stimulus in one context ([30%, 50%]; in terms of how likely it was to receive a reward) but was the worse stimulus in the other context ([50%, 70%]). However, different from Experiments 1 and 2, 30% and 70% were closer to 50% compared with 10% and 90%. This experiment thus preserved the better/worse asymmetry for the 50% reward stimuli but allowed us to examine whether context effect on 50% would change as a function of its distance in reward probability from the other stimuli.

The results of Experiment 5 were mixed. Consistent with our model prediction, for 50% reward, subjects gave larger probability estimate (red in Fig 13B) in the context in which the other stimulus carried a smaller reward probability (30%) than in the context where the other stimulus carried a larger reward probability (70%; blue in Fig 13B). Also consistent with our model prediction was that the size of the context effect on 50% reward was smaller compared with Experiments 1 and 2. However, inconsistent with our model prediction, for both 30% and 70%, probability estimates were virtually identical between different contexts toward the end of the session (Figs 10A and 13C, respectively). Finally, we found that subjects' choice behavior in the lottery decision task was consistent with their probability estimates (Fig 13D).

In summary, Experiments 3 to 5 allowed us to examine in more detail the impact of uncertainty and reference-dependent computations on probability estimation and largely replicated the effects observed in Experiments 1 and 2. First, we found that reference dependency acts strongly in driving context effect especially for stimulus whose reward probability was in the middle of the three probabilities tested in each experiment. This is similar to the reference-dependent gain/loss asymmetry observed in outcome evaluation [1,11] and indicates that people estimate reward probability relative to some reference point—the average reward probability experienced in the recent past. Second, we found that uncertainty plays a key role in driving context effect because regardless of its position among the three probabilities tested in different experiments, there was always—although to varying degree—context effect on 50% probability estimate. The same cannot be said for probabilities—30% and 70%—whose uncertainty was smaller than 50%. Third, in Experiment 5 ([30%, 50%, 70%]), we found that probability estimates at both 30% and 70% were virtually identical between different contexts. This suggests that the no-effect on 10% and 90% in Experiments 1 and 2 were less likely because of these probabilities reaching the boundary on the probability scale. However, this finding was

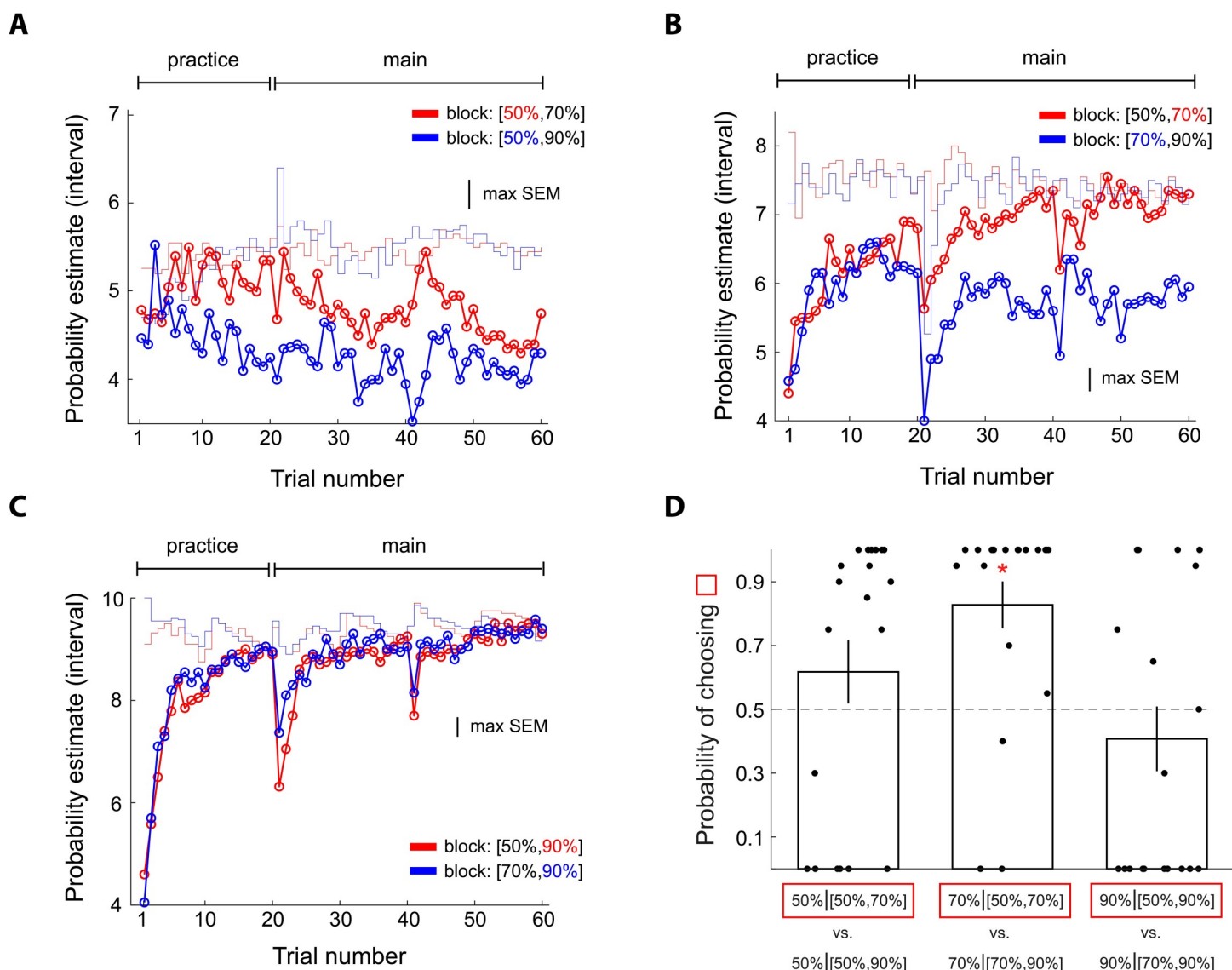

**Fig 12. Experiment 4: Examining context effects on probability estimates at 50%, 70%, and 90% reward and finding further support for the URD model.**
Conventions are the same as in Fig 5. The design principle of the experiment was identical to the design of Experiments 1 and 2 (Fig 1B) except that subjects faced 50%, 70%, and 90% reward probabilities instead of 10%, 50%, and 90% reward. (A) Probability estimates of 50% reward. (B) Probability estimates of 70% reward. (C) Probability estimates of 90% reward. (D) Choice probability in the lottery decision task after the probability estimation task. Error bars represent ± 1 SEM. The red star symbol indicates a significant difference from 0.5 based on 95% bootstrap confidence interval. Data underlying these graphs can be found in https://osf.io/48j7m. URD, uncertainty and reference dependent.

also inconsistent with our model prediction—because there was a moderate level of uncertainty at both 30% and 70%, the URD model predicts a greater difference in probability estimate than what was actually observed. This implies that the impact of uncertainty is smaller than expected and that compared with reference point, the impact of uncertainty might be smaller. Nonetheless, the findings across these 5 experiments further solidify uncertainty dependency and reference dependency as two computational building blocks for context effects on probability estimation.

Finally, in Experiment 6 ($N = 20$ subjects), we tested whether monetary incentives would change the context effects observed in Experiments 1 and 2. Subjects were told that they would

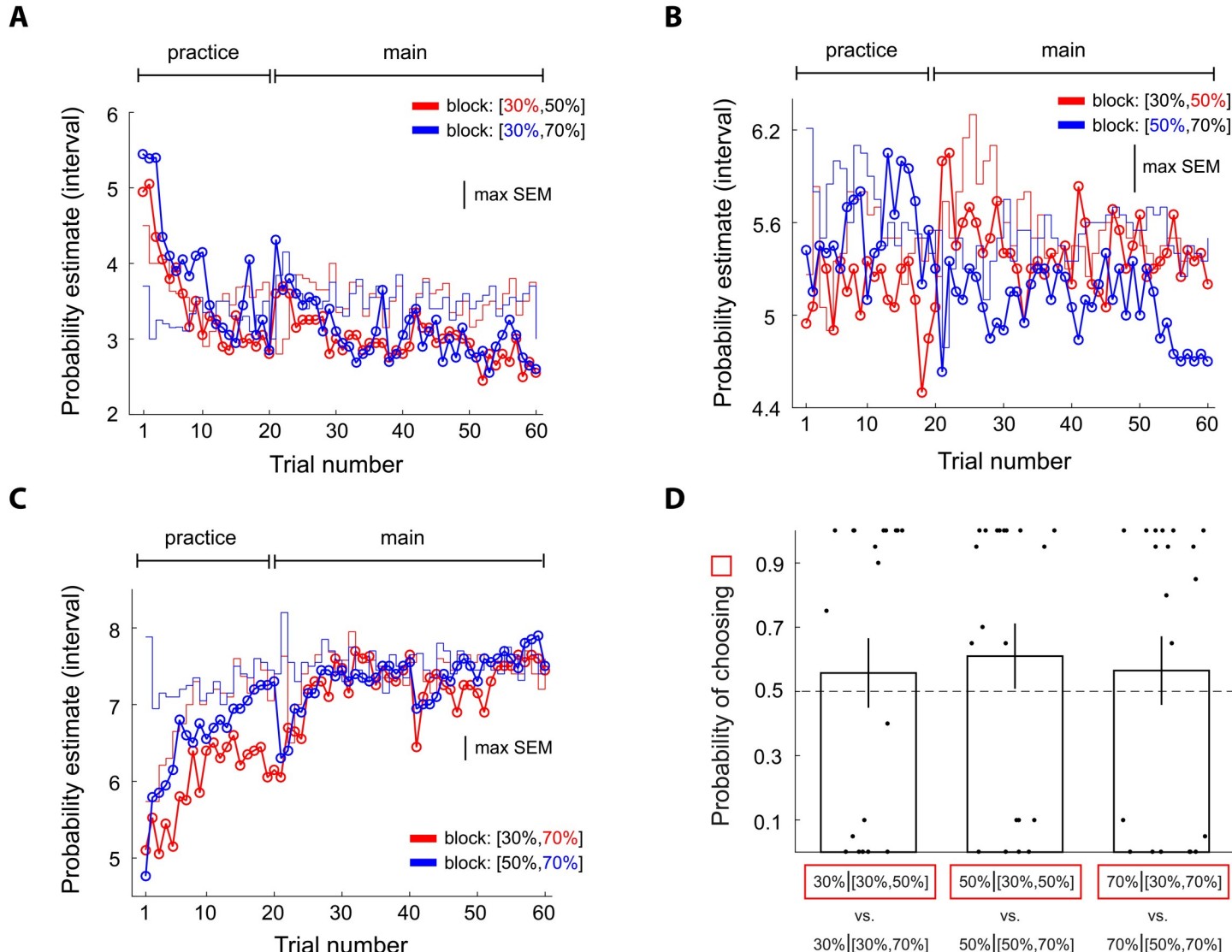

**Fig 13. Experiment 5: Examining context effects on probability estimates at 30%, 50%, and 70% reward and finding further support for the URD model.** Conventions are the same as in Fig 5. The design principle of the experiment was identical to the design of Experiments 1 and 2 (Fig 1B) except that subjects faced 30%, 50%, and 70% reward probabilities instead of 10%, 50%, and 90% reward. (A) Probability estimates of 30% reward. (B) Probability estimates of 50% reward. (C) Probability estimates of 70% reward. (D) Choice probability in the lottery decision task after the probability estimation task. Error bars represent ± 1 SEM. Data underlying these graphs can be found in https://osf.io/48j7m. URD, uncertainty and reference dependent.

receive additional monetary bonus on a trial-by-trial basis based on how accurate his or her probability estimates were to the true probabilities. During the experiment, subjects were not given feedback on how accurate they were and the monetary bonus. Instead, they were told that their probability estimates were recorded and that at the end of the experiment they would receive the sum of additional bonus based on their trial-by-trial accuracy. We replicated our findings in Experiments 1 and 2 by showing significant context effect on 50% reward probability but not 10% and 90% (Fig 14A–14C). Further, context effect on 50% probability estimates was stronger than both 10% or 90%, which was also reflected in subjects' choice behavior in the lottery decision task following the probability estimation task (Fig 14D).

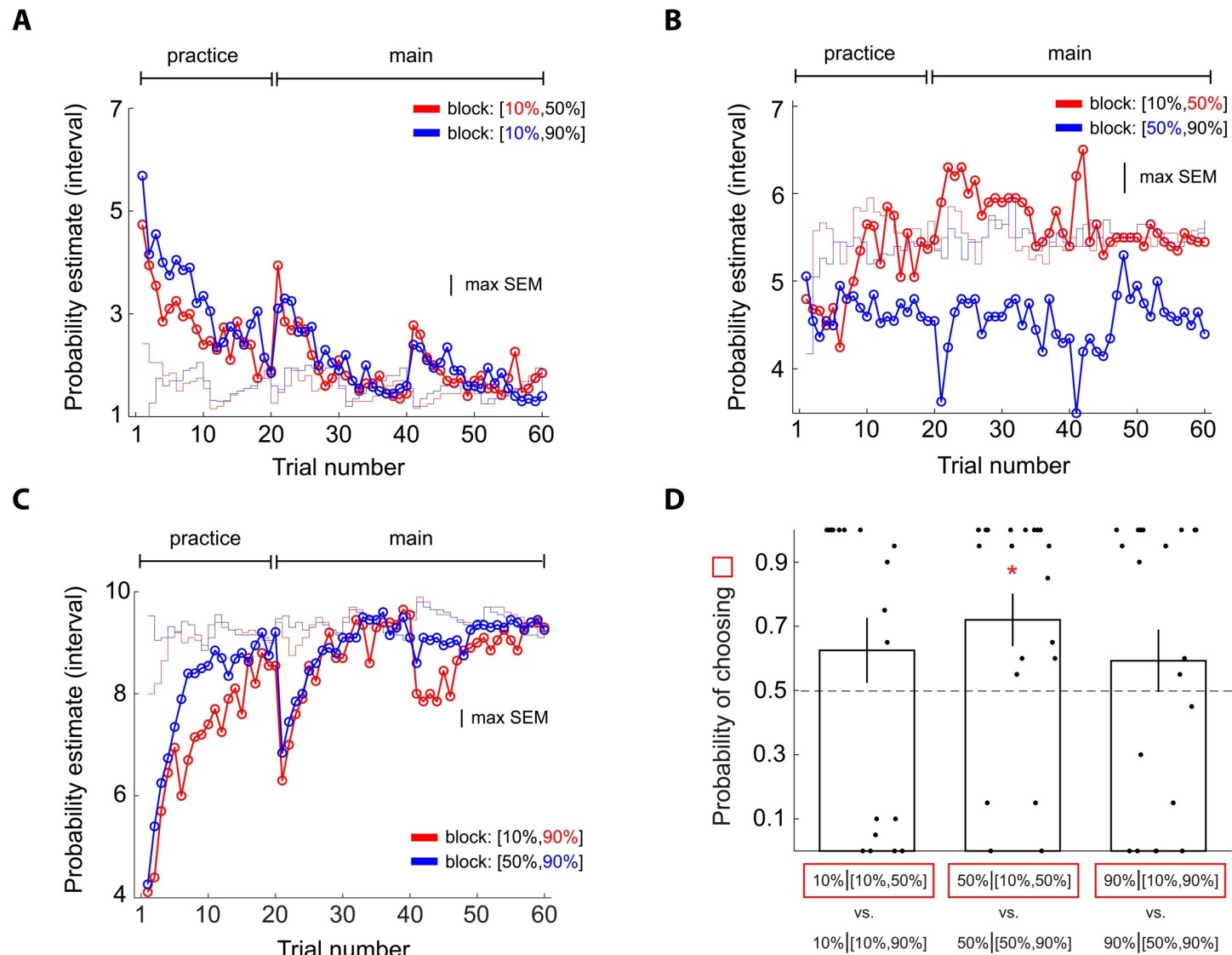

**Fig 14. Experiment 6: Using an incentive-compatible design to examine context effects on probability estimates at 10%, 50%, and 90% reward and replicating the context effects seen in Experiments 1 and 2.** Conventions are the same as in Fig 5. The design principle of the experiment was identical to the design of Experiments 1 and 2 (Fig 1B) except that we provided monetary incentives to the subjects to give accurate probability estimates. (A) Probability estimates of 10% reward. (B) Probability estimates of 50% reward. (C). Probability estimates of 90% reward. (D) Choice probability in the lottery decision task after the probability estimation task. Error bars represent ± 1 SEM. The red star symbol indicates a significant difference from 0.5 based on 95% bootstrap confidence interval. Data underlying these graphs can be found in https://osf.io/48j7m.

## Discussion

This study was motivated by two observations on decision making under uncertainty. First, estimating probability of uncertain events is necessary to making good choices. In uncertain environments, events are probabilistic in nature and therefore having access to probability information is critical. In most cases, the decision maker is not given complete information about probability. Organisms must therefore estimate probability of events that are carriers of value in order to make good and adaptive choices. Second, context of experience should impact probability estimation. Because we rarely experience an event in isolation, the context of our experience—the presence of other events unfolding close in time—should play an

important role in probability estimation. Here, we found robust context effects on probability estimates: when a stimulus carried intermediate probabilities of reward (e.g., 50/50 of reward or no reward), subjects overestimated or underestimated its reward probability depending on whether the other stimulus present in the same context carried a smaller (e.g., 10%) or larger probability of reward (e.g., 90%), respectively. By contrast, subjects did not show significant context effect at extreme probabilities (e.g., 10% or 90% reward).

## Computational building blocks for context-dependent probability estimation

We developed a mathematical model for context-dependent probability estimation and tested whether it can describe the behavior observed in this study. In developing our model, we hypothesized that two computational building blocks—reference dependency and uncertainty dependency—play important roles in probability estimation. The first building block, reference dependency, describes that when estimating the probability of reward associated with a stimulus, the average probability of all stimuli a decision maker encounters in a context would serve as a reference point for comparison. People tend to overestimate reward probability if the average probability is smaller than the stimulus reward probability. In contrast, if the average probability is larger than the stimulus probability, people tend to underestimate its probability of reward. Reference dependency has long been suggested in the evaluation of outcomes, treating outcomes as gains or losses relative to some reference point [1], which has been formally modeled as the decision maker's expectation of future outcomes based on recent experience [2,48]. However, there is very little work on examining whether probability estimation is reference-dependent [49]. Conceptually, the notion of reference point is similar to base rate (e.g., [50]). In the Bayesian framework, the predicted impact of base rate is such that probability estimates would be biased towards the base rate. By contrast, in the reference-dependency framework, probability estimates would be biased away from the reference point. Therefore, these two frameworks are conceptually distinct and would make opposite predictions.

The second building block, uncertainty dependency, should only exist in estimation problems that are uncertain in nature. Given that there is more uncertainty regarding which potential outcome would occur when estimating probabilities that are near 0.5, they are more susceptible to context effect. This uncertainty dependency predicts that the reference-point effect on probability estimation would be modulated by the degree of uncertainty, i.e., variance or standard deviation of experienced outcomes associated with the stimulus of interest such that the effect is stronger at intermediate probabilities (larger uncertainty) than extreme probabilities (smaller uncertainty). These predictions are consistent with what we found in the current study: when stimulus reward probability was at the extremes (10% or 90% chance of reward), context did not significantly impact subjects' probability estimates. By contrast, when a stimulus carried a 50% reward, subjects overestimated reward probability when the average reward probability in a context was smaller than 50% but underestimated reward probability when the average reward probability was larger than 50%. These results argue against a purely categorical model in which probabilities are not coded numerically but in discrete categories with ordinal relationships. In contrast to our model, such a categorical model cannot account for the magnitude of the context effect at different probabilities observed in this study.

The model we developed in this study and reinforcement learning models [44,51,52] are not mutually exclusive. In fact, we incorporated all three model classes—our URD model, DN, and RN—into the Rescorla–Wagner reinforcement-learning model framework and fit them to individual subjects' probability-estimate data. Compared with fits in the non-Rescorla–Wagner model framework (past outcomes are equally weighted to compute frequency statistics),

the Rescorla–Wagner model fits tended to capture the trial-to-trial dynamics better, suggesting that decaying impact of outcomes into the past better captures the reward frequency statistics used by the decision maker when estimating probability of reward.

## Comparison between context-dependent probability estimation and subjective-value computations

Although this study was inspired by previous research that examined context effect on valuation, it differs from previous work in that we asked whether and how context impacts probability estimation. Our work demonstrates similarities in that probability estimation is, like subjective-value estimation [2,48,53–63], context dependent. However, the results also highlight significant differences that are likely due to challenges imposed by estimating probability that are unique to valuation. Recent neurobiological research also has begun to investigate how context impacts subjective-value representations in the brain [6,28,29,31,32,39,40,64,65] and how context-dependent value signals relate to choice behavior [8,66].

Here, we discuss both similarities and differences between probability estimation and valuation, from the perspective of the tasks, computations, and algorithms carried out by the decision maker. First, although reward is present in both, a notable difference between valuation and probability estimation tasks is the presence or absence of uncertainty. The majority of valuation studies examined value computations in the absence of uncertainty (e.g., a stimulus is always associated with 3 drops of apple juice). In contrast, in our task subjects had to estimate the probability of receiving a reward associated with visual stimuli. Another difference in tasks is that although stimulus-reward associations are present in both, subjects' own preferences for different rewards can clearly impact the subjective value of a stimulus but not its probability of reward. Instead, probability of an event such as reward is often determined by the outside world rather than subjective preferences that are internal in nature. Second, from the perspective of computations, by focusing on identifying computational building blocks for probability estimation and using them to guide model development, we were able to conclude that reference dependency is one crucial building block for both probability estimation and subjective-value computations. However, uncertainty dependency is unique to estimating rewards when there are uncertainties involved. Third, from the perspective of algorithms, our results suggest that there are perhaps more similarities between probability estimation and subjective-value computations than differences. The model comparison statistics showed that our context-dependent model (URD model) did not differ significantly from normalization models that have been useful in describing subjective-value computations. However, a key difference between these models lies at the large end of the extreme probabilities: at 90% reward, our model did not predict context effect—consistent with subjects' behavior—but normalization models did. Our model offers a simple explanation to the lack of context effect here. That is, because the impact of reference point on probability estimates is scaled by the degree of uncertainty, the impact of reference point would be very small at 90% reward in which subjects almost always received a reward and hence there is very small uncertainty on whether she or he would receive or not receive a reward.

In contrast to our investigation on probability estimation, Palminteri and colleagues [30] focused on value representations—whether they are context-dependent during learning. They manipulated valence—using monetary gains and losses—of potential outcomes associated with different lotteries under consideration in a decision-making task in which subjects had to learn different options through outcome feedback after each choice. They showed that a relative, context-dependent reinforcement-learning model that incorporates reference dependency in the computation of option value better explained subjects' behavioral data and brain

activity in key valuation areas in VMPFC and striatum than the absolute reinforcement-learning model in which value computations are context-independent. Although our study focused on probability estimation and theirs on valuation, the computational models we separately developed shared a key feature in reference dependency. Together, our results highlight the importance of expectation as reference point in both value and probability computations. It should be noted that reference-dependent preference is an active area of research in behavioral and experimental economics in which expectation-based models for reference-dependent preferences have been proposed [2,48,67–70] and tested [61–63,71–74]. These models, including ours and in the work by Palminteri and colleagues [30], share the feature of expectations as reference point, but they also differ in some other key aspects, for example, the definition of expectation and how it is updated. Therefore, it is important in future work to investigate similarities and differences between these models and compare how well they describe behavioral and neural data on valuation and probability estimation.

## Neural representations for context effects on probability estimation

Combining univariate and multivariate analyses, our fMRI results indicate that the neural implementations for probability estimation involve a network of brain regions, with dACC, VMPFC, and IPS representing context effects on probability estimates and VS representing the average reward frequency statistic necessary for context-dependent probability estimation. The VMPFC finding, together with previous studies showing its involvement in context-dependent valuation [6,29,30,75], established VMPFC as the common neurocomputational substrate for probability estimation and valuation. By contrast, the involvement of dACC may reflect aspects of probability estimation that is unique to valuation, which has been indicated in tracking uncertainty-related statistics, such as variance and volatility [17,21], comparing recent and distant reward history [22,23], reinforcement-guided learning [76], and decision making under uncertainty [77]. Similarly, the IPS had also been shown to represent valuation signals in decision under ambiguity [34] and decision under uncertainty [33].

A potential alternative interpretation of our dACC finding on representing context effect on 50% reward probability estimates is associated with response conflict or conflict monitoring [78–81] that also showed strong dACC involvement. Recall that in our fMRI experiment (Experiment 1), we assigned two buttons to represent the probability intervals close to 50%—one for the 35% to 50% interval and the other for the 50% to 65% interval. This forced the subjects to choose one of the two competing buttons—therefore initiating a response conflict—when indicating his or her probability estimates of the stimuli whose reward frequency was close to 50%.

We, however, believe that response conflict is less likely to explain our dACC finding for the following reasons. First, for the 50% reward stimuli, response conflict should be present in both the [10%, 50%] and [50%, 90%] contexts. And yet our fMRI MVPA analysis used the difference in brain activity in response to 50% stimuli between the two contexts. Therefore, if dACC represents response conflict that was present in both contexts, then the difference in dACC activity between these two contexts should not reflect response conflict. Hence, the dACC MVPA results we observed are perhaps less likely to be caused by response conflict. Second, the response conflict hypothesis does not specifically provide prediction on the direction and magnitude of context effect, i.e., 50% estimate is smaller in the [50%, 90%] context than the [10%, 50%] context, which we found dACC to represent by tracking individual differences in such behavioral metric. Therefore, even though dACC had been shown to represent conflict and the detection/monitoring of conflict, its activity in the context of our experiment cannot be directly attributed to response conflict or conflict detection/monitoring.

### Probability estimation and experience-based decision making

By showing that probability estimation is context-dependent, our results provide insights into experience-based decision making—an area of research that investigates decision making where choosers acquire information about different options through experience—in that context-dependent probability estimation is another source of bias that could affect people's choice behavior. This research had shown that people nonlinearly weight probability information learned through experience [15] even when their probability estimates matched well with objective probabilities [82]. However, context was typically not manipulated in these studies, and therefore it is not possible to examine how context-induced bias on probability estimation would affect choice. It would be interesting to examine how the presence of these two sources of bias—probability weighting and context-dependent probability estimation—would impact choice behavior. At the neural algorithmic level, representations of probability distortion had been shown in the striatum [83], lateral prefrontal cortex [20], and VMPFC [84]. Given that our results indicate that dACC, VMPFC, IPS (MVPA results), and striatum (average reward frequency) are critical to context effect, it remains open as to whether these two sources of biases are represented in the same brain regions and how they are being combined to impact experience-based decisions. In the future, it would also be important to examine, along with the stimulus standard deviation statistic, how other sources of variability in the environment could contribute to context effect. For example, how would the randomness or the rate of switching between different stimuli presented to the subjects influence context effects on probability estimates? How might this source of variability interact with the stimulus standard-deviation statistic to impact probability estimation?

In many choices we face, we are not explicitly given information about potential outcomes and their associated probabilities of occurrence. This makes probability estimation an essential computation for decision making under uncertainty. Finding that probability estimation is context-dependent not only demonstrates that it is subjective but, more importantly, that it is relative. Such relative subjective probability is represented in brain regions thought to be involved in subjective-value computations (VMPFC) and regions involved in extracting reward statistics from the environments in order to make decisions under uncertainty (dACC and IPS).

Probability estimation is a general problem organisms face in uncertain environments. Indeed, estimating probability through experience is an essential component of computational accounts of a wide array of cognitive tasks—from making sensory inferences, predicting future outcomes, to choosing between uncertain prospects [37,38,85]. Statistically optimal models provide formal accounts of cognition that require probability estimation of various events but has often failed to consider how context, namely, the presence of different events unfolding close in time, could impact probability estimation of each event of interest [86]. Thus, the robust context effects we observed and the computational building blocks identified suggest that context needs to be considered for the many cognitive computations that require estimation of probability.

## Materials and methods

### Ethics statement

All subjects gave written informed consent prior to participation in accordance with the procedures approved by the Taipei Veteran General Hospital IRB. The experiments were conducted according to the principles expressed in the Declaration of Helsinki.

### Subjects

A total of 137 subjects (67 women, mean age = 25.3 years) participated in this study. Among them, 37 right-handed subjects participated in the fMRI experiment (Experiment 1); three

subjects did not complete the pre-fMRI session. To be consistent in data analysis across subjects, these three subjects' behavioral and fMRI data were not analyzed. Each of the remaining 5 experiments was purely behavioral (Experiments 2 to 6) and had 20 subjects each. For the fMRI experiment, the subjects received 500 NTD (1 US Dollar = 30 New Taiwan Dollar or NTD) for their participation and received an additional bonus from the experiment. Their average earning was 1,120 NTD. For behavioral experiments, subjects received 225 NTD for their participation. The average additional bonus was 504 NTD (Experiment 2), 395 NTD (Experiment 3), 651 NTD (Experiment 4), 612 NTD (Experiment 5), and 581 NTD (Experiment 6). The difference in bonuses across experiments likely reflected the difference in reward probabilities used in these experiments.

## Task

We designed a simple stimulus-outcome association task to investigate how context affects probability estimation. Below, we describe the trial sequence of the task and our manipulation of context. The task was programmed using the Psychophysics Toolbox in MATLAB [87,88]. Visual stimuli were projected from a LCD projector outside the scanner and viewed through a mirror mounted on the head coil. Responses were collected via two MRI-compatible button boxes each consisting of 4 buttons.

**Trial sequence.** On each trial, a red dot was first presented for 0.5 s to indicate the start of a trial. This was followed by the presentation of a visual stimulus. Prior to the experiment, the subjects were instructed that each stimulus she or he faced carried a certain probability of reward that she or he had no knowledge of. Upon seeing the stimulus, subjects had up to 2 s to indicate his or her estimate of its reward probability with a button press. Failure to do so would prevent the lottery to be executed, and as a result, she or he won nothing in the current trial. There were 8 possible buttons each corresponding to a particular interval of probabilities: [0%–5%], [5%–20%], [20%–35%], [35%–50%], [50%–65%], [65%–80%], [80%–95%], [95%–100%]. Each button was assigned a finger to press. Subjects had to select the interval that best described his or her estimate of probability of reward associated with the current stimulus. Two opposite types of button assignments were implemented—left-to-right or right-to-right—and were balanced across subjects. In the left-to-right assignment, the probability increased from the left-most finger (left pinkie) to the right-most finger (right pinkie). The thumbs were not assigned. This design served to rule out confounds related to motor preparation and execution that would correlate with reward probability if not properly controlled. Once a button was pressed, subjects received visual feedback (0.5 s) on the probability interval he or she had indicated with the pressed button. This feedback only served to confirm the subject's choice and did not give any information about the veridical probability of reward. A variable delay (1–7 s) then followed before feedback on reward was provided (2 s). The receipt or no receipt of a reward was determined by sampling with replacement from reward probability of the stimulus presented on the trial. If there was a reward, then the amount of reward was randomly selected (1 NTD to 5 NTD in steps of 1 NTD). In other words, reward magnitude was independent of reward probability. These design rules were explicitly stated to the subjects prior to the experiment. After the reward/no-reward feedback, a variable intertrial interval was presented (1–9 s) before the next trial began. There was no incentivization for being accurate in probability estimates—subjects did not receive additional monetary bonus if probability estimates were close to the true reward probabilities (but see Experiment 6 in subsection "Four additional behavioral experiments (Experiments 3 to 6)" below for an incentivized version of the task).

**Manipulation of context.** The subjects experienced three different contexts (context 1, 2, and 3) in the stimulus-outcome association task. Context was implemented in a block of trials

by placing two visual stimuli in it (Fig 1B). The order of their presentation was randomized for each block separately. Each stimulus in the context was assigned a unique probability of reward. In each trial, the subjects saw one of the two stimuli and, through reward feedback, learned information about its reward probability. Three reward probabilities, 10%, 50%, and 90%, were implemented in the task. Each probability appeared in two different contexts and was represented by two different visual stimuli. For instance, in both contexts 1 and 3, one of the two stimuli carried a 50% chance of reward. However, the reward probability of the other stimulus was different between these two contexts. In context 1, the other stimulus carried a 10% chance of reward. By contrast, in context 3, the other stimulus had a 90% chance of reward. This design therefore allowed us to compare two stimuli carrying the same probability of reward but experienced in two different contexts. Specifically, we asked the questions—how would context affect probability estimates? How might the effect of context change as a function of reward probability?

## Procedure

The fMRI experiment (Experiment 1) consisted of 3 distinct phases—pre-fMRI, fMRI, and post-fMRI sessions. The experiment took approximately 80 min to complete.

**Pre-fMRI session.** Before the fMRI session, subjects first completed a behavioral session of the stimulus-outcome association task before entering the MRI scanner. There were three randomly ordered context blocks (contexts 1, 2, and 3 mentioned in the subsection "Task" above) of 40 trials each. In each block, there were 20 trials for each visual stimulus. The order of the two stimuli presented within each block was randomized. This session served to train subjects to perform the task and to establish a certain degree of knowledge through experience about the reward probability of each stimulus under different contexts.

**fMRI session.** There were six blocks of trials in the session, two for each context. The ordering of the blocks was randomized for each subject separately. In each block, there were 30 trials, 15 trials for each of the two possible visual stimuli. The order of the two stimuli presented within each block was randomized.

**Post-fMRI session.** After the fMRI session, subjects performed a lottery decision task in which they were asked to choose between visual stimuli she or he faced in the previous two sessions. At this point, subjects should have acquired a certain degree of knowledge about the reward probability of each stimulus through experience. In each trial, the subjects were presented with two different visual stimuli. She or he was instructed that the amount of reward associated with winning was the same at 250 NTD across all stimuli and that she or he should choose the one preferred based on the experience with the stimuli, i.e., subjective belief about probability of reward. The subjects were told that at the end of the session one of his or her choices would be selected at random and realized. Because 2 visual stimuli were selected out of 6 stimuli in each trial, there were 15 possible pairs of visual stimuli. Among them, 3 pairs had stimuli with the same reward probability that were experienced under different contexts. For example, one pair had two visual stimuli both with 10% chance of reward. The 10% stimulus in context 1 was experienced along with a 50% stimulus. This 10% stimulus is referred to as the (10% | [10%, 50%]) stimulus. In contrast, the 10% stimulus in context 2 was experienced along with a 90% stimulus. This 10% stimulus is referred to as the (10% | [10%, 90%]) stimulus. Besides the (10% | [10%, 50%]) versus (10% | [10%, 90%]) pair, there were also the (50% | [10%, 50%]) versus (50% | [50%, 90%]) pair, and the (90% | [10%, 90%]) versus (90% | [50%, 90%]) stimuli. The subjects' choices in these 3 pairs thus allowed us to examine how context affected choice under each probability (10%, 50%, and 90% reward) and compare context effect between these probabilities. In principle, there were 20 trials for each of these 3 pairs, and 7 trials for the each of the remaining 12 pairs. Hence, there were a total of 144 trials.

However, because of a programming error, not all pairs had the same number of trials: they ranged from 3 to 43 trials across subjects. The average number of trials for the 10% reward, 50% reward, and 90% reward pairs was 27 (standard deviation = 7.64), 17 (standard deviation = 11.86), and 24 (standard deviation = 9.05), respectively. We corrected this error in the behavioral control experiment (*n* = 20 trials, for each pair on each subject; see subsection "Behavioral control experiment (Experiment 2)" below) in which we replicated the choice result and also in all subsequent experiments (see subsection "Four additional behavioral experiments (Experiments 3 to 6)" below).

## Behavioral control experiment (Experiment 2)

One potential limit of Experiment 1 was how we assigned buttons to the intervals of reward probability. The button that included 10% corresponded to the interval between 5% and 20%, the button that included 90% corresponded to the interval between 80% and 95%. It is possible that context also played a role in subjects' probability estimates on the 10% and 90% stimuli, but as long as the difference in probability estimates between different contexts was not large enough such that estimates in the two contexts corresponded to two different buttons, we would not be able to detect it. In contrast, we had a more sensitive measure for context effect on the 50% probability stimuli, because there were two buttons around 50%—one corresponding to the interval between 35% and 50% and the interval between 50% and 65%.

To rule out this potential confound, we conducted a behavioral control experiment by changing the button-to-probability assignment. In this experiment (*N* = 20 subjects), there were 10 keys, each corresponding to a 10% interval—[0%–10%], [10%–20%], [20%–30%], [30%–40%], [40%–50%], [50%–60%], [60%–70%], [70%–80%], [80%–90%], and [90%–100%]. For each stimulus, there were a total of 40 trials in the main session, which was equivalent to the fMRI session in Experiment 1 that had 30 trials for each stimulus. Both the interstimulus interval (time gap between stimulus presentation and reward back) and the intertrial interval were 1 s long. In addition, for all reward probabilities, the frequencies of reward each subject experienced matched with their corresponding objective probabilities of reward. This served to control the variability in frequency of reward (across subjects and across contexts for the same probability within a subject) caused by sampling with replacement. Otherwise, the control experiment was identical to Experiment 1.

## Four additional behavioral experiments (Experiments 3 to 6)

The design of Experiments 3 to 6 was identical to Experiment 2 except that the reward probabilities tested were different. In Experiment 3, we tested 10%, 30%, and 50% reward in three different contexts: [10%, 30%], [10%, 50%], and [30%, 50%]. In Experiment 4, we tested 50%, 70%, and 90% reward in three different contexts: [50%, 70%], [70%, 90%], and [50%, 90%]. In Experiment 5, we tested 30%, 50%, and 70% reward in three different contexts: [30%, 50%], [30%, 70%], and [50%, 70%]. In Experiments 5 and 6, we included additional monetary bonus when subjects gave accurate probability estimate. The probabilities tested in Experiment 6 were the same as in Experiments 1 and 2. We designed the additional bonus based on the following rule: if subjects pressed a button whose interval overlapped with ±5% of the true reward probability, they would receive a 1 NTD bonus. For example, if the true reward probability was 50%, then subjects would receive 1 NTD bonus if she or he pressed either the 40% to 50% or 50% to 60% button. If probability was 70%, subjects would receive 1 NTD if she or he pressed either the 60% to 70% or 70% to 80% button. Subjects were told that they would not receive feedback on additional bonus during the experiment and that at the end of the experiment they would receive the sum of additional bonus they earned during the experiment.

**fMRI data acquisition.** Imaging data were collected with a 3T MRI scanner (Siemens Magnetom Skyra) equipped with a 32-channel head array coil in the Taiwan Mind and Brain Imaging Center at National Chengchi University. T2*-weighted functional images were collected using an EPI sequence (TR = 2 s, TE = 30 ms, 35 oblique slices acquired in ascending interleaved order, 3.5 × 3.5 × 3.5 mm isotropic voxels, 64 × 64 matrix in a 224-mm field of view, flip angle 90˚). Each subject completed 6 EPI runs in the fMRI session. Each run consisted of 210 images. T1-weighted anatomical images were collected after the EPI runs using an MPRAGE sequence (TR = 2.53 s, TE = 3.3 ms, flip angle = 7˚, 192 sagittal slices, 1 × 1 × 1 mm isotropic voxel, 256 × 256 matrix in a 256-mm field of view). For each subject, field map image was also acquired for the purpose of estimating and partially compensating for geometric distortion of the EPI images so as to improve registration performance.

**fMRI preprocessing.** The following preprocessing steps were applied using FMRIB Software Library (FSL) [89]. First, motion correction was applied using MCFLIRT to remove the effect of head motion during each run. Second, FUGUE (FMRIB's Utility for Geometrically Unwarping EPIs) was used to estimate and partially compensate for geometric distortion of the EPI images using field map images collected for the subject. Third, spatial smoothing was applied using a Gaussian kernel with either FWHM = 8 mm (GLM-1) or FWHM = 5 mm (GLM-2). Fourth, a high-pass temporal filtering was applied using Gaussian-weighted least square straight line fitting with $\sigma$ = 50 *s*. Fifth, registration was performed in a 2-step procedure. First, the unsmoothed EPI image that was the midpoint of a run was used to estimate the transformation matrix (7-parameter affine transformations) from EPI images to the subject's high-resolution T1-weighted structural image, with nonbrain structures removed using FSL's Brain Extraction Tool (BET). Second, transformation matrix (12-parameter affine transformation) from the high-resolution T1-weighted structural image to the standard MNI template brain was estimated.

## General linear modeling of BOLD response

**GLM-1.** At the time of stimulus presentation, each visual stimulus (there were two in a block) was modeled by an indicator regressor whose length on each trial was the subject's response time (RT) to indicate probability estimate. At the time of reward feedback, an indicator regressor, a regressor for the RPE, a regressor for the magnitude of reward outcome (MAG), and the interaction between RPE and MAG were implemented. All regressors were convolved with a canonical gamma hemodynamic response function. Temporal derivatives of each regressor were included in the model as regressors of no interest.

**Contrasts for examining context effect at stimulus presentation.** We estimated the BOLD response to each stimulus and compared BOLD response between stimuli carrying the same reward probability but were experienced in different contexts. For each probability, at the individual-subject level, we compared the BOLD response between two different contexts in a fixed-effect model. The resulting beta estimates were then fed into a group-level linear mixed-effect model to analyze whether there was a significant difference at the group level.

**Contrasts for RPE and MAG.** After first-level time-series analysis for each block separately, beta estimates for RPE and MAG were entered into a linear fixed-effect model for each subject. The results of the fixed-effect model on RPE and MAG were separately entered into a group-level mixed-effect model to analyze the effect of RPE and MAG at the group level.

**Covariate analysis on context effect.** This analysis focused on the 50% reward stimuli that showed significant context effect on probability estimates. At the subject level, we separately estimated the BOLD response magnitude to the 50% reward stimulus in the [10%, 50%] context ($\beta_{[10\%,50\%]}^{50\%}$) and in the [50%, 90%] context ($\beta_{[50\%,90\%]}^{50\%}$). For each subject, we computed

the difference between $\beta_{[10\%,50\%]}^{50\%}$ and $\beta_{[50\%,90\%]}^{50\%}$ ($\beta_{[10\%,50\%]}^{50\%} - \beta_{[50\%,90\%]}^{50\%}$), referred to as $\Delta\beta_{50\%}$. We then performed a group-level covariate analysis on $\Delta\beta_{50\%}$ (linear mixed-effect model) by using $\Delta\hat{P}_{50\%}$—the difference in the subject's mean probability estimate (averaged across trials) on the 50% reward stimulus—as a covariate.

**GLM-2.** At stimulus presentation, we implemented three regressors, an indicator regressor and two parametric regressors. The length of these regressors is the subject's trial-by-trial RT. One parametric regressor was the frequency of reward associated with the visual stimulus presented on the current trial ($f_S$) and the other parametric regressor was the frequency of reward collapsed across different stimuli ($f_{overall}$). Both were computed based on reward history in the past 10 trials. At the time of feedback, the regressors were identical to GLM-1. All regressors were convolved with a canonical gamma hemodynamic response function. Temporal derivatives of each regressor were included in the model as regressors of no interest.

**Contrasts for $f_S$ and $f_{overall}$.** The analysis steps were identical to the one described in the subsection "Contrasts for RPE and MAG" above.

**Functional ROIs.** Five functional ROIs—the dACC, VMPFC, VS, bilateral aINS, and bilateral IPS—were used for MVPA analysis. The dACC, aINS, and IPS ROIs were created using term-based meta-analyses available in neurosynth.org. The term for dACC ROI was dorsal anterior and we used the mask based on association test. The aINS and IPS ROIs (bilateral) were created using the conjunction map of two terms—uncertainty and probability—and used the mask based on uniformity test. For dACC, we excluded voxels that are inconsistent with anatomical labeling of the cingulate cortex. The resulting dACC ROI contained 1,350 voxels (2 mm isotropic). The VMPFC ROI (1,838 voxels, 2 mm isotropic) was created based on the work by Barta and colleagues [26] that showed significant activation at the outcome receipt stage. The VS ROI (246 voxels, 2 mm isotropic) was based on the work by Clithero and Rangel [27] that significantly correlated with subjective value. The left aINS ROI contained 272 voxels, right aINS ROI contained 344 voxels, left IPS ROI contained 196 voxels, and right IPS ROI contained 312 voxels.

**LOSO ROI analysis.** We created independent and unbiased ROIs in the VS using LOSO method based on the RPE contrast described in GLM-1. The LOSO method was used to examine RPE representations (Fig 10B), $f_S$ and $f_{overall}$ representations (Fig 10C) in the VS. For each subject separately, we performed the analysis in the following steps. First, we identified regions that significantly correlated with RPE using all other subjects' data (cluster-level inference using cluster-forming threshold z > 2.3, familywise error corrected at $p < 0.05$). Second, we identified the voxel with the maximum z-statistic in the VS cluster that significantly correlated with RPE and created a sphere mask centered at this voxel (radius = 6 mm). Third, we computed the mean beta value of the contrast of interest (RPE, $f_S$ or $f_{overall}$) across voxels in the mask. After obtaining mean beta value for each subject separately, we performed a one-sample $t$ test on the mean beta (across subjects) against 0 at the $\alpha = 0.05$ level.

**Between-subject MVPA.** The purpose of this analysis was to use MVPA analysis to identify neural representations for individual differences in context effect on probability estimates. Because context effect was significant when stimuli carried 50% reward, the analysis focused on these stimuli. The behavioral measure of context effect is $\Delta\hat{P}_{50\%}$—the difference in mean probability estimate (across trials) when 50% reward stimulus was in the [10%, 50%] context and in the [50%, 90%] block—which we computed for each subject separately. At the neural level, for each subject separately, we obtained an $\Delta\beta_{50\%}$ image as described in the subsection "Covariate analysis on context effect" above under GLM-1. We adopted a searchlight procedure [90] that allows the extraction of information from local pattern of multivoxel brain activity. We focused on the five ROIs—VMPFC, VS, dACC, bilateral aINS, and bilateral IPS—

described above (see the subsection "Functional ROIs"). For a given voxel, we defined a spherical mask centered on it (radius = 8 mm). We used $\Delta\hat{P}_{50\%}$ to label each of the multidimensional pattern vectors (vectors of multivoxel $\Delta\beta_{50\%}$) and performed an $n$-fold ($n$ = 33) cross-validation to predict the normalized $\Delta\hat{P}_{50\%}$. We excluded one subject from the analysis because she or he had 1/3 missing trials—trials in which no probability estimate was provided by the subject—for the 50% stimuli. Therefore, we performed the analysis on 33 subjects. We also performed the analysis by including all subjects and did not find the results to differ significantly (S5 Fig). On each cross-validation run, we trained a linear SVR machine with the labeled data from 32 subjects and tested on the independent data of the remaining subject. The SVR was performed using the LIBSVM implementation (http://www.csie.ntu.edu.tw/~cjlin/libsvm) with a linear kernel and a constant regularization parameter of c = 1. Prior to SVR, both the continuous label ($\Delta\hat{P}_{50\%}$) and the multidimensional fMRI pattern vector for a given voxel were normalized across subjects according to $x_{norm} = \frac{x - \min(x)}{\max(x) - \min(x)}$, where $x$ is the value before normalization, $x_{norm}$ is the normalized value, $\max(x)$ is the maximum value of $x$, and $\min(x)$ is the minimum value of $x$. On each cross-validation run, the normalization parameters ($\min(x)$, $\max(x)$) were computed based on the training data set and then applied to both the training and test data set. This was to prevent the possibility that dependencies between test and training data set were introduced. For each voxel separately, we obtained 33 predicted $\Delta\hat{P}_{50\%}$ based on the above procedure. Prediction performance was determined by calculating the Pearson correlation coefficient between the predicted and the actual $\Delta\hat{P}_{50\%}$.

To assess whether prediction performance was significantly above chance level, we performed nonparametric permutation tests. For each voxel separately, we repeatedly trained and tested the SVR by randomly permuting the continuous label ($\Delta\hat{P}_{50\%}$) in order to generate a null distribution of the correlation coefficients. Prediction accuracy was considered significant if the probability of the true correlation occurring was $p < 0.05$, Bonferroni-corrected for multiple comparisons based on the number of voxels tested in a priori ROIs in the dACC (1,350 voxels), VMPFC (1,838 voxels), VS (246 voxels), left aNIS (272 voxels), right aINS (344 voxels), left IPS (196 voxels), and right IPS (312 voxels). A whole-brain mask obtained from the univariate GLM analysis of this data set was applied to these ROIs so that the voxels tested in these ROIs contained data from all subjects. As a result, the number of tested voxels in dACC, VMPFC, VS, left aINS, right aINS, left IPS, and right IPS was 1,350; 1,776; 202; 272; 344; 196; and 312, respectively. The number of permutations run for each ROI was $1 \div (0.05 \div n)$, where $n$ is the number of tested voxels. Therefore, the number of permutations ($n_{permutations}$) run was 27,000; 35,520; 4,040; 5,440; 6,880; 3,920; and 6,240 for the dACC, VMPFC, VS, left aINS, right aINS, left IPS, and right IPS ROIs, respectively. If the true correlation coefficient exceeded the largest value of the created null-distribution of correlation coefficients, the nonparametric $p$-value was reported as being smaller than the lowest possible nonparametric $p$-value given the number of permutations ($1/n_{permutations}$). Note that even though the total number of possible permutations is fixed, this number is extremely large, which makes performing the analysis on all possible permutations very computationally expensive. This is why we used the approach described above. In this case, the number of significant voxels could vary slightly each time one runs permutation test on the voxels within an ROI. However, the results reported survive even after we repeatedly performed permutation testing according to the procedure described above.

**Modeling context-dependent computations for probability estimation: The URD model.** We developed a computational model—the URD model—that takes into account context and reward history for probability estimation. The subject's probability estimate ($\hat{P}_S$)

depends on two terms. First, it depends on the frequency of reward experienced in the recent past ($f_S$). Second, it depends on a context term, which is the difference between $f_S$ and the overall frequency of reward ($f_{overall}$) experienced in the recent past. Critically, the impact of the context term is determined by a parameter, which we refer to as the susceptibility parameter ($\tau$). This is equivalent to saying that there is a gain-control mechanism that regulates the impact of reference-dependent computation ($f_S$–$f_{overall}$) and that the susceptibility parameter represents the gain factor. The model is expressed by the following equation:

$$\hat{P}_S = f_S + \tau(f_S - f_{overall}).\tag{2}$$

We assume that $\tau$ is sensitive to the estimated uncertainty on which potential outcome would occur, which can be defined by the estimated variance ($\hat{\sigma}_S^2$) or standard deviation of experienced outcomes associated with the stimulus ($\hat{\sigma}_S$). Note that in the model-fitting exercise (see the subsection "Model fitting and model comparison" below) we assumed $\tau$ to be $\hat{\sigma}_S$ or $\hat{\sigma}_S^2$—special cases for $\tau$ to be monotonically increasing as a function of either $\hat{\sigma}_S$ or $\hat{\sigma}_S^2$, which we assumed.

In this study, subjects received binary reward outcomes (a reward or no reward). Hence, $\hat{\sigma}_S$ would be largest when the stimulus carried a 50% chance of reward. The model thus makes the prediction that the subjects' probability estimates of the 50% reward stimulus are affected the most by ($f_S$–$f_{overall}$) than the other stimuli that carried either 10% reward or 90% reward. The model also predicts that when a 50% reward stimulus was experienced with a 10% reward stimulus, subjects would overestimate reward probability, as ($f_S$–$f_{overall}$) would be greater than 0. In contrast, when the 50% reward stimulus was experienced with a 90% reward stimulus, subjects would underestimate reward probability because ($f_S$–$f_{overall}$)<0.

To directly test the model, we can implement $f_S$ and ($f_S$–$f_{overall}$) in a multiple regression analysis. Alternatively, we can just use $f_S$ and $f_{overall}$ as the two regressors in the regression model. This model has an advantage in that it reduced colinearity between the two regressors. Therefore, for each subject and each reward probability separately, we performed the following regression analysis:

$$\hat{P}_S(t) = \beta_{f_S} f_S(t; n_{past}) + \beta_{f_{overall}} f_{overall}(t; n_{past}),\tag{3}$$

where $\hat{P}_S(t)$ is subjects' estimate of reward probability associated with the stimulus presented in trial $t$, $f_S(t; n_{past})$ is the reward frequency of the stimulus in trial $t$ calculated based on the past $n_{past}$ trials and $f_{overall}(t; n_{past})$ is the overall reward frequency (the average reward frequency across stimuli in a context) on trial $t$ calculated based on the past $n_{past}$ trials.

To make the computational model directly comparable to the regression model, we rewrite Eq 2:

$$\hat{P}_S = (1 + \tau)f_S - \tau f_{overall}.\tag{4}$$

The model thus makes the following predictions on the regression coefficients: First, $\hat{\beta}_{f_S}$ should be positively and significantly different from 0 across all reward probabilities. Second, $\hat{\beta}_{f_S}$ associated with 50% reward stimuli should be larger than that of the 10% reward and 90% reward stimuli. Third, $\hat{\beta}_{f_{overall}}$ associated with the 50% reward stimuli should be negative and its magnitude should be larger than that of the 10% reward and 90% reward stimuli. Finally, $\hat{\beta}_{f_S}$ should be equal to $(1 - \hat{\beta}_{f_{overall}})$. For each subject and each reward probability separately, we performed the regression analysis in Eq 3 with $n_{past}$ ranging from 1 to 30 (the number of trials in a block). We then selected the $n_{past}$ that best described the data for each subject and each reward

probability separately and used the corresponding regression coefficients for further analyses (Fig 6B and 6C).

## Model fitting and model comparison

**Computational models.** A total of 9 different models—three versions of the URD model, four versions of the DN model, and two versions of the RN model—were fitted to the subjects' average probability estimates and were compared using model comparison statistic (BIC). We also fitted data at the individual-subject level using the same models described below. In addition, we incorporated these models into the Rescorla–Wagner reinforcement-learning model framework and fitted them at the individual-subject level (S1 Text).

The URD model takes the following form,

$$\hat{P}_S = f_S + \hat{\sigma}_S(f_S - f_{overall}), 0 \leq P_S \leq 1, \tag{5}$$

as described in Eq 2, and we explicitly modeled $\tau$ by the estimated standard deviation of past reward outcomes ($\tau = \hat{\sigma}_S$). We also fitted versions of the same model with $\tau = \hat{\sigma}_S^2$ (S4 Fig). When fitting the group average data, we used the past 30 trials (the length of a block) to calculate $f_s, f_{overall}, \hat{\sigma}_S^2, \hat{\sigma}_S$. In general, BIC values tended to be very similar after 5 past trials (S3 Fig). When fitting individual-subject data, we fitted each model using different number of past trials (from 1 to 30), selected the number that best described the data and use it to compute the model fits on probability estimates and the BIC values.

To further model the overestimation of small probabilities and underestimation of large probabilities, which we observed in our data set and was also found in the work by Fox and Tversky [13], we used the following equation to model these estimation biases:

$$w_P = \frac{f_S^\gamma}{(f_S^\gamma + (1 - f_S)^\gamma)^{1/\gamma}}, \gamma > 0, \tag{6}$$

where $w_P$ represents the distorted probability estimate and $\gamma$ the free parameter that captures the nonlinear distortion of probability estimate. Note that when $0 < \gamma < 1$, small $f_S$ are overestimated and moderate to large $f_S$ are underestimated. Then the probability estimate is computed by the same principles:

$$\hat{P}_S = w_p + \hat{\sigma}_S(w_p - f_{overall}), 0 \leq P_S \leq 1. \tag{7}$$

Eq 7 is referred to as URD-$\gamma$. Finally, in URD-$\gamma$–$\lambda$, we modeled loss aversion [1,11] by updating Eq 7:

$$\hat{P}_S = \begin{cases} w_P + \hat{\sigma}_S(w_P - f_{overall}), \text{if } w_P - f_{overall} \geq 0 \\ w_P + \lambda\hat{\sigma}_S(w_P - f_{overall}), \text{if } w_P - f_{overall} < 0 \end{cases}, \tag{8}$$

where $\lambda$ represents the loss-aversion parameter and is strictly greater than 0.

For divisive normalization models, we fit two forms. The first form (DN-1) [32] is

$$\hat{P}_S = \frac{a + f_S}{b + f_S + f_{OS}}, \tag{9}$$

where $f_{OS}$ is the reward frequency of the other stimulus (OS) present in the same context as $f_S$, and $a$ and $b$ are the free parameters. For example, in the [10%, 50%] context, if $f_S$ corresponds to the stimulus carrying 50% chance of reward, then $f_{OS}$ would be the stimulus carrying 10%

reward. The second form (DN-2) [91] is

$$\hat{P}_S = \frac{af_S}{1 + bf_{overall}}. \qquad (10)$$

For both DN-1 and DN-2, we also considered probability weighting such that $\hat{P}_S$ was transformed using Eq 6 to become the final probability estimate (DN-1-$\gamma$,DN-2-$\gamma$).

The RN model [32] takes the following form:

$$\hat{P}_S = \frac{a + f_S}{b + \max([f_S, f_{OS}]) - \min([f_S, f_{OS}])}, 0 \leq P_S \leq 1, \qquad (11)$$

where $a$ and $b$ are the free parameters. We also considered probability weighting such that $\hat{P}_S$ was transformed using Eq 6 to become the final probability estimate (RN-$\gamma$). In Table 1, we summarize the 9 different models and their respective free parameters. Note that the parameter $\sigma_{\text{noise}}$ (Eq 12) is not considered a free parameter of these models. Rather, it is the free parameter for modeling the stochasticity of probability estimate $\hat{P}_S$ once it is being computed by these models.

These models weight past outcomes equally when computing the reward frequency and variance statistics and hence are different from the Rescorla–Wagner reinforcement-learning model framework. We therefore referred to them as the non-Rescorla–Wagner models. We used subjects' probability-estimate data in the fMRI session when fitting non-Rescorla–Wagner models (for both group and individual-subject fitting). When fitting models in the Rescorla–Wagner model framework, we incorporated data from both pre-fMRI and fMRI sessions. We wish to point out that the conclusion of the model fits and model comparison would be the same regardless of whether to include data from pre-fMRI session. Note that for the individual-subject model fitting (both Rescorla–Wagner and non-Rescorla–Wagner model frameworks), in models that required computing either reward frequency statistic or variance statistic, or both, the results shown in Fig 8 and S4 Fig are based on the number of past trials, for each model separately, that produced the largest value of maximum likelihood across different numbers of past trials (from 1 to 30).

**Maximum-likelihood estimation.** We fit the computational models described above to the time-series data of subjects' probability estimates (for fitting group average data, we used the mean probability estimates averaged across all subjects; for fitting individual-subject data, we used the subject's trial-by-trial probability estimates). We did not include trials in which subjects did not give probability estimate in our fitting. We assumed that probability estimate is a Gaussian random variable with mean at $\hat{P}_S^{\text{model}}$ and variance $\sigma_{\text{noise}}^2$, where $\hat{P}_S^{\text{model}}$ is the

**Table 1. Models and their free parameters.**

| Model class | Model name | Free parameters |
|---|---|---|
| URD models | URD | none |
| | URD-$\gamma$ | $\gamma$ |
| | URD-$\gamma$–$\lambda$ | $\gamma, \lambda$ |
| DN models | DN-1, DN-2 | $a, b$ |
| | DN-1-$\gamma$,DN-2-$\gamma$ | a, b, $\gamma$ |
| RN models | RN | $a, b$ |
| | RN-$\gamma$ | a, b, $\gamma$ |

**Abbreviations:** DN, divisive normalization; RN, range normalization; URD, uncertainty and reference dependent

model-predicted probability estimate and $\sigma^2_{\text{noise}}$ is a free parameter. The likelihood function is therefore

$$L(\hat{P}^{\text{actual}}_S; \Theta_{\text{model}}, \sigma^2_{\text{noise}}) = \prod_i^n \frac{1}{\sqrt{2\pi\sigma^2_{\text{noise}}}} e^{-\frac{(\hat{P}^{\text{actual}}_S(i) - \hat{P}^{\text{model}}_S(i))^2}{2\sigma^2_{\text{noise}}}}, \tag{12}$$

where $\hat{P}^{\text{actual}}_S$ is a vector containing the subjects' probability estimates, $\Theta_{\text{model}}$ represents the set of free parameters in a model, and $n$ is the number of total trials evaluated. For example, in the URD model (Table 1), $\Theta_{\text{model}}$ is empty because there is no free parameter in it, and therefore there is only one free parameter $\sigma^2_{\text{noise}}$ to be estimated when fitting URD.

**Model comparison.** When fitting the group average data, for each model separately, we used a nonparametric bootstrap method to reconstruct the distribution of BIC described below,

$$\text{BIC} = \ln(n)k - 2\ln(L_{\text{max}}), \tag{13}$$

where $n$ represents the number of trials used when fitting a model, $k$ represents the number of free parameters (number of parameters in Table 1 plus one for $\sigma^2_{\text{noise}}$) in the model, and $L_{\text{max}}$ represents the value of maximum likelihood given by the best-fitting parameters. To statistically compare different models using BIC, we performed nonparametric bootstrapping to estimate the confidence interval of BIC for each model separately. That is, we resampled from the subject pool with replacement 10,000 times to construct 10,000 resampled data set. For each resampled data set, we computed the mean probability estimate (across subjects) associated with each trial in the order. We then fitted the probability estimates with different models and based on the fitting result computed their corresponding BIC. As a result, for each model separately, we obtained a distribution of BIC values, which we used to compute the 95% confidence interval that allowed us to statistically compare between different models.

**Nonparametric bootstrap test.** We used nonparametric bootstrap procedure to construct confidence interval for the statistic of interest and test for significance [41] at the $\alpha = 0.05$ level. To do this, we resampled with replacement from all the subjects to construct a resampled data set and use it to compute the statistic of interest (e.g., mean choice probability). We then repeated this procedure for 10,000 times so as to construct the 95% confidence interval of the statistic.

## Supporting information

**S1 Fig. Experiment 1: RT data.** Here, we plot the dynamics of mean RT (across subjects) over the course of the experiment (pre-fMRI session and fMRI session) separately for each reward probability in each context. (A) 10% reward. (B) 50% reward. (C) 90% reward. Conventions are the same as Fig 4. fMRI, functional magnetic resonance imaging; RT, response time (TIF)

**S2 Fig. Experiment 1: Examining the consistency of context effect across different subgroups of subjects.** It is possible that context effect on the 50% probability estimates (Fig 4B) was driven by reward frequency bias in the pre-fMRI session, with 50% reward in the [10%, 50%] context (red) having larger reward frequency than that in the [50%, 90%] context (blue). To address this issue, we divided subjects into three subgroups according to their experience in the pre-fMRI session and plotted average probability estimates for each subgroup separately. We found context effect consistent with Fig 4B across all three subgroups. (A) Subgroup 1 (14 subjects): Subjects who experienced smaller reward frequency when facing the 50% reward stimulus in the [10%, 50%] context than the 50% reward stimulus in the [50%, 90%] context in the pre-fMRI session. (B) Subgroup 2 (6 subjects): Subjects who experienced the same reward

frequency when facing the 50% reward stimuli between [10%, 50%] and [50%, 90%] contexts. (C) Subgroup 3 (14 subjects): Subjects who experienced larger reward frequency when facing the 50% reward stimulus in the [10%, 50%] context than the 50% reward stimulus in the [50%, 90%] context. We found that the subjects in all three subgroups showed the context effect consistent with Fig 4B. That is, regardless of the subjects' experience in the pre-fMRI session, for 50% reward, they gave larger probability estimates in the [10%, 50%] context (red) than in the [50%, 90%] context (blue). This suggests that the context effect shown in Fig 4B is not because of bias in reward frequency the subjects experienced in the pre-fMRI session. fMRI, functional magnetic resonance imaging
(TIF)

**S1 Text. Fitting computational models for probability estimation in the Rescorla–Wagner reinforcement-learning model framework.** Here, we describe in detail how we fit different computational models for probability estimation in the Rescorla–Wagner reinforcement-learning model framework.
(DOCX)

**S3 Fig. Model fitting: Issue on number of past trials.** In the paper, we fit different models to subjects' probability estimates. In one class of models (the non-Rescorla–Wagner model framework), we considered a time window into the past—namely, the number of past trials—when calculating the reward frequency and variance statistics used by the models. In this case, the number of past trials became a free parameter. We found that the model fits—in BIC values—decreased as a function of window length and would vary little after 5 trials into the past. Here, we show BIC values based on fitting group average data plotted against number of past trials considered to compute frequency statistics for model computations. (A) The URD models (9 versions). (B) DN models (6 versions). (C) RN models (4 versions). Model abbreviations are the same as in the main text. In addition, UOS is for a version of URD in which the frequency of overall reward is replaced by the reward frequency of the OS, and URDv is a version of the URD in which the estimated standard deviation of potential outcomes is replaced by the estimated variance of potential outcomes. URD version 2 (ver_2) is a version of URD in which both reward frequency of the stimulus of interest and the overall reward frequency are transformed into probability weight based on the weighting function with free parameter $\gamma$ (gamma; Eq 6 in "Materials and methods") when computing the probability estimate. BIC, Bayesian information criterion; DN, divisive normalization; OS, the other stimulus present in the same context as the stimulus of interest; RN, range normalization; URD, uncertainty and reference dependent.
(TIF)

**S4 Fig. Model-fitting results.** As described in the main text, we fit computational models at both the group and individual-subject levels. When fitting individual-subject data, we considered two model frameworks—the Rescorla–Wagner and the non-Rescorla–Wagner model frameworks. Here, we show model fits from the non-Rescorla–Wagner framework (A–B) and the BICs from both frameworks (C–D). (A–B) Fitting results from the non-Rescorla–Wagner framework. (A) Model fits of DN-1, DN-2, and RN. (B) Model fits of DN-1-$\gamma$, DN-2-$\gamma$, and RN-$\gamma$. (C–D) Model comparison based on BIC. (C) Non-Rescorla–Wagner framework (23 models). (D) Rescorla–Wagner framework (22 models, without RN-1param due to poor convergence). Model abbreviations are the same as in S3 Fig. In addition, version 3 (ver_3) of URD is a version of URD in which loss aversion is performed first before probability weighting, _wSigma indicates a version of URD in which a free weighting parameter is multiplied with estimated uncertainty (either the standard deviation or variance of potential outcomes) in

the URD computation, and _wGain is a version of URD in which a free weighting parameter is multiplied with the reference-dependent term $\hat{\sigma}_S(w_p - f_{overall})$ in Eq 8 in Materials and methods when $w_p - f_{overall} > 0$. BIC, Bayesian information criterion; DN, divisive normalization; RN, range normalization; URD, uncertainty and reference dependent.
(TIF)

**S5 Fig. MVPA analysis.** Conventions are the same as in Fig 9 in the main text. In Fig 9, we present MVPA results that excluded one subject's data because she or he had too many missing trials (she or he did not provide probability estimate within two seconds after stimulus onset in 1/3 of the 50% reward trials), making the estimates of BOLD response less reliable compared with other subjects. Here, we show results from including this subject's data in the analysis. As expected, the results are not identical to those shown in Fig 9. However, they are similar in the sense that dACC—at both stimulus presentation and reward feedback—represented individual subjects' context effect on probability estimates (S5A, S5B and S5C Fig), VMPFC represented individual subjects' context effect on probability estimates at the time of reward feedback (S5D and S5E Fig), and right IPS represented individual subjects' context effect on probability estimates at the time of stimulus presentation (S5F Fig). BOLD, Blood oxygen level dependent; dACC, dorsal anterior cingulate cortex; IPS, intraparietal sulcus; MVPA, multivoxel pattern analysis; VMPFC, ventromedial prefrontal cortex
(TIF)

**S6 Fig. ROI analysis on probability-estimate representations in VMPFC and VS.** To investigate neural representations for probability estimates, we ran a GLM identical to GLM-1 (see the subsection "General linear modeling of BOLD response" in "Materials and methods") with the exception of adding a parametric regressor representing subjects' trial-by-trial probability estimates at the time of stimulus presentation. At the whole-brain level, we did not find regions that significantly correlated with probability estimates. We performed ROI analysis in the VMPFC and VS based on previous meta-analysis papers in value-based decision making and also did not find these ROIs to represent trial-by-trial probability estimates. The ROIs used were identical to those shown in Fig 10 in the main text. Here, we show results from using sphere masks (radius = 8 mm) centered at the peak coordinates for subjective value in VMPFC ([x−2, y40, z−6]) and VS ([x−8, y8, z−6]) identified in Clithero and Rangel. The mean beta value was not significantly different from 0 in both ROIs (VS: t = −0.788, df = 33, $p$ = 0.437; VMPFC: t = 0.468, df = 33, $p$ = 0.643). We also used masks from Bartra and colleagues and did not see the beta value of probability estimate to differ significantly from 0 (VS: t = −0.91, df = 33, $p$ = 0.37; VMPFC: t = −0.08, df = 33, $p$ = 0.936). In summary, we did not find VMPFC and VS to represent subjects' trial-by-trial probability estimates at the time of stimulus presentation. GLM, general linear model; ROI, region of interest; VMPFC, ventromedial prefrontal cortex; VS, ventral striatum
(TIF)

**S7 Fig. PPI analysis.** In this study, we found that dACC represented individual differences in context effect on probability estimates based on MVPA analysis (Fig 9). To further examine whether dACC showed task-related functional connectivity with regions shown to represent reward statistics, namely, VS for representing the overall frequency of reward associated with a particular context, we performed the following PPI analysis using dACC as the seed region (sphere mask with 8 mm radius centered at [x2, y30, z18]—the voxel with the strongest effect in MVPA analysis). The PPI model implemented two PPI contrasts, one for the interaction between the dACC time series and the onset regressor at the time of stimulus presentation and the other for the interaction between seed time series and onset regressor at the time of reward

feedback. These two contrasts allowed us to examine regions that show changes in functional connectivity with dACC at the time of stimulus presentation and at the time of reward feedback separately. The rest of the regressors in the PPI model were identical to GLM-1 (see the subsection "General linear modeling of BOLD response" in "Materials and methods"). We performed ROI analysis in VMPFC and VS based on previous meta-analysis papers in value-based decision making and found that dACC did not show changes in functional connectivity with VS at both time windows but showed decrease in functional connectivity with VMPFC at the time of stimulus presentation. The ROIs used were identical to those shown in Fig 10. (A) ROI analysis on PPI contrast at the time of stimulus presentation. The beta value represents the regression coefficient of the PPI contrast. (Left two bars) VS and VMPFC ROIs from Clithero and Rangel: VS: t = −0.327, df = 33, $p$ = 0.746; VMPFC: t = −2.094, df = 33, $p$ = 0.044. (Right two bars) VS and VMPFC ROIs from Bartra and colleagues: VS: t = −0.335, df = 33, $p$ = 0.74; VMPFC: t = −1.93, df = 33, $p$ = 0.063. (B) ROI analysis on PPI contrast at the time of reward feedback. The beta value represents the regression coefficient of the PPI contrast. (Left two bars) VS and VMPFC ROIs from Clithero and Rangel: vStr: t = 0.038, df = 33, $p$ = 0.97; VMPFC: t = −0.463, df = 33, $p$ = 0.646. (Right two bars) VS and VMPFC ROIs from Bartra and colleagues: VS: t = −0.05, df = 33, $p$ = 0.96; VMPFC: t = −0.752, df = 33, $p$ = 0.457. In summary, we did not find dACC to show significant changes in functional connectivity with VS that represents the overall frequency of reward at both time windows. We did, however, find significant decrease in functional connectivity between dACC and VMPFC at the time of stimulus presentation. dACC, dorsal anterior cingulate cortex; GLM, general linear model; PPI, Psycho-physiologic interaction; MVPA, multivoxel pattern analysis; ROI, region of interest; VMPFC, ventromedial prefrontal cortex; VS, ventral striatum
(TIF)

**S1 Table. Reward-magnitude representations.** Cluster-level inference was performed (familywise error corrected at $p < 0.05$) using Gaussian random field theory with a cluster-forming threshold $p < 0.001$ (z > 3.1).
(DOCX)

**S2 Table. Reward-magnitude representations.** We performed nonparametric permutation test using the TFCE option in randomise (FSL) and performed 5,000 permutations. The $p$-value represents the familywise error corrected $p$-value. FSL, FMRIB software library; TFCE, threshold-free cluster enhancement.
(DOCX)

**S3 Table. Prediction-error representations.** We performed nonparametric permutation test using the TFCE option in randomise (FSL) and performed 5,000 permutations. The $p$-value represents the familywise error corrected $p$-value. FSL, FMRIB software library; TFCE, threshold-free cluster enhancement.
(DOCX)

**S4 Table. Stimulus reward-frequency representations.** We performed nonparametric permutation test using the TFCE option in randomise (FSL) and performed 5,000 permutations. The $p$-value represents the familywise error corrected $p$-value. FSL, FMRIB software library; TFCE, threshold-free cluster enhancement.
(DOCX)

## Acknowledgments

We thank Wan-Yu Shih, Chia-Jen Lee and Yi-Ju Liu for their help on data collection.

## Author Contributions

**Conceptualization:** Wei-Hsiang Lin, Shih-Wei Wu.

**Data curation:** Wei-Hsiang Lin, Shih-Wei Wu.

**Formal analysis:** Wei-Hsiang Lin, Justin L. Gardner, Shih-Wei Wu.

**Funding acquisition:** Shih-Wei Wu.

**Investigation:** Wei-Hsiang Lin, Justin L. Gardner, Shih-Wei Wu.

**Methodology:** Wei-Hsiang Lin, Justin L. Gardner, Shih-Wei Wu.

**Resources:** Shih-Wei Wu.

**Software:** Justin L. Gardner, Shih-Wei Wu.

**Supervision:** Shih-Wei Wu.

**Validation:** Wei-Hsiang Lin, Justin L. Gardner, Shih-Wei Wu.

**Visualization:** Wei-Hsiang Lin, Justin L. Gardner, Shih-Wei Wu.

**Writing – original draft:** Wei-Hsiang Lin, Justin L. Gardner, Shih-Wei Wu.

**Writing – review & editing:** Wei-Hsiang Lin, Justin L. Gardner, Shih-Wei Wu.

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
