## [Editor Report · Decision Letter 0]

5 Aug 2019

Dear Dr Wu, 

Thank you for submitting your manuscript entitled "Context effect on probability estimation" for consideration as a Research Article by PLOS Biology.

Your manuscript has now been evaluated by the PLOS Biology editorial staff, as well as by an Academic Editor with relevant expertise, and I am writing to let you know that we would like to send your submission out for external peer review.

*Please be aware that, due to the voluntary nature of our reviewers and academic editors, manuscripts may be subject to delays during the holiday season. Thank you for your patience.*

Please re-submit your manuscript within two working days, i.e. by Aug 07 2019 11:59PM.

Kind regards,

Gabriel Gasque, Ph.D.,

Senior Editor

PLOS Biology

---

## [Decision Letter · Decision Letter 1]

13 Sep 2019

Dear Dr Wu,

Thank you very much for submitting your manuscript "Context effect on probability estimation" for consideration as a Research Article at PLOS Biology. Your manuscript has been evaluated by the PLOS Biology editors, by an Academic Editor with relevant expertise, and by three independent reviewers.

In light of the reviews (below), we will not be able to accept the current version of the manuscript, but we would welcome resubmission of a much-revised version that takes into account the reviewers' comments. We cannot make any decision about publication until we have seen the revised manuscript and your response to the reviewers' comments. Your revised manuscript is also likely to be sent for further evaluation by the reviewers.

Your revisions should address the specific points made by each reviewer. As you will see, they have requested a series of additional analyses that we think, together with the Academic Editor, you should perform. In addition, reviewer 1 and reviewer 3 think that extra data collection might be needed to strengthen your conclusions. We have also discussed these recommendations with the Academic Editor and think that for a successful revision, you should provide these additional data. 

Please submit a file detailing your responses to the editorial requests and a point-by-point response to all of the reviewers' comments that indicates the changes you have made to the manuscript. In addition to a clean copy of the manuscript, please upload a 'track-changes' version of your manuscript that specifies the edits made. This should be uploaded as a "Related" file type. You should also cite any additional relevant literature that has been published since the original submission and mention any additional citations in your response. 

Before you revise your manuscript, please review the following PLOS policy and formatting requirements checklist PDF: http://journals.plos.org/plosbiology/s/file?id=9411/plos-biology-formatting-checklist.pdf. It is helpful if you format your revision according to our requirements - should your paper subsequently be accepted, this will save time at the acceptance stage.

Please note that as a condition of publication PLOS' data policy (http://journals.plos.org/plosbiology/s/data-availability) requires that you make available all data used to draw the conclusions arrived at in your manuscript. If you have not already done so, you must include any data used in your manuscript either in appropriate repositories, within the body of the manuscript, or as supporting information (N.B. this includes any numerical values that were used to generate graphs, histograms etc.). For an example see here: http://www.plosbiology.org/article/info%3Adoi%2F10.1371%2Fjournal.pbio.1001908#s5.

For manuscripts submitted on or after 1st July 2019, we require the original, uncropped and minimally adjusted images supporting all blot and gel results reported in an article's figures or Supporting Information files. We will require these files before a manuscript can be accepted so please prepare them now, if you have not already uploaded them. Please carefully read our guidelines for how to prepare and upload this data: https://journals.plos.org/plosbiology/s/figures#loc-blot-and-gel-reporting-requirements.

Upon resubmission, the editors will assess your revision and if the editors and Academic Editor feel that the revised manuscript remains appropriate for the journal, we will send the manuscript for re-review. We aim to consult the same Academic Editor and reviewers for revised manuscripts but may consult others if needed.

We expect to receive your revised manuscript within two months. Please email us (plosbiology@plos.org) to discuss this if you have any questions or concerns, or would like to request an extension. At this stage, your manuscript remains formally under active consideration at our journal; please notify us by email if you do not wish to submit a revision and instead wish to pursue publication elsewhere, so that we may end consideration of the manuscript at PLOS Biology.

When you are ready to submit a revised version of your manuscript, please go to https://www.editorialmanager.com/pbiology/ and log in as an Author. Click the link labelled 'Submissions Needing Revision' where you will find your submission record. 

Sincerely,

Gabriel Gasque, Ph.D., 

Senior Editor

PLOS Biology

Reviewer remarks:

Reviewer #1: This paper looks at context effects in probability learning. They find that, particularly for uncertain options, a context of higher probability (compared to lower) alternatives decreases the perceived probability (and vice versa). This is a relatively new area in neuroscience in which there has not been that much work. However, one notable study – Palminteri et al. (2015) (‘contextual modulation of value signals in reward and punishment learning’) - is quite similar in key aspects: the study also shows that probabilities are perceived differently depending on the context of the other alternatives being present and identified vmPFC and dmPFC/dACC as related to this phenomenon. What is conceptually new here is that the authors show that contextual effects are stronger for stimuli about which there is more uncertainty (though potentially an additional control experiment may be needed to rule out alternative explanations). Furthermore, the present study is very thoroughly done; in particular it is noteworthy that first, the behavioural results are replicated in a second sample. Second, results are shown both in a model-based and model-free framework, meaning that small arguments about how the models were specified do not affect the general value of the finding. Third, results are shown in terms of ratings as well as choices post-task, highlighting the robustness of the effect.

Major comments:

Introduction/Discussion

1. Palminteri et al. (2015) is only mentioned in passing. However, this study actually appears to be quite similar to the present one – both in terms of behavioural and neural effects - and that should be acknowledged and discussed

Results - behaviour

2. The authors highlight the role of uncertainty to explain the differences between the conditions where they find strongest effects (50%) and where they don’t (10% and 90%). However, it is also the case that the most certain options (i.e. 10% and 90%) are at the same time also those that are on the boundary of the rating scales. An alternative approach would be to use ‘noisy magnitudes’ (e.g. a stimulus is associated with reward drawn from a standard deviation with mean 50 points and with a standard deviation of 10 points). In this case, noisiness could be manipulated without approaching the boundary of the rating scale. If the authors want to reinforce the point about uncertainty, I would recommend collecting an additional behavioural sample with this manipulation of the task. 

3. Modelling: the authors use as comparison model one adapted from Louie et al. (divisive normalization). I have several questions about this: 

a. Why does this model show an asymmetry in what it predicts for the 10% and 90% stimuli? 

b. It seems that the authors have adapted the equation – i.e. they use Ps=(a+fs)/(b+fs+fOS), rather than in e.g. Khaw et al. (2017) P=a*fs/(1+b*fOverall). Why? 

c. I could not quite figure this out, but it might be relevant to consider literature on the ‘base rate’ phenomenon in a Bayesian framework (e.g. see Koehler (1996)). I have wondered whether participants somehow might think that an outcome is the result of a stimulus and the base rate of outcomes in that context. However, this is only a suggestion and authors can ignore this.

4. Modelling: it appears from the methods (section Model fitting and model comparison> Maximum-likelihood estimation) that instead of fitting each person’s data, the model was fit on the average across participants. This is highly unusual and there is some data to suggest that this approach does not capture the real behaviour that well (Ahn et al., 2013). I would suggest that the authors either fit each person separately or that they use a hierarchical framework (e.g. can be done in the free software ‘Stan’, see Carpenter et al. (2017). These models should then be also used for model comparison. 

5. Modelling: why do you use fs/foverall as a non-weighted sum of past trials, rather than the more typical Rescoral-Wagner weighted effect? Can you show by replacing e.g. fs in the regression by the past trial outcomes that in fact they are weighted equally?

6. Modelling: referring to table 4 (list of free parameters in models), it appears that the divisive normalization models did not allow for gamma (as in equation 6), i.e. over/underweighting of probabilities. By looking at figure 6, it appears that the uncertainty models particularly provide a better model fit when this gamma (or ‘wp’ in figure 6) is included. What happens if it is also included in the range normalization or divisize normalization models, are these then in fact the best performing models? I am asking this as it appears that in terms of BIC (so already correcting for the number of free parameters quite stringently), the range normalization/divisive normalization models are very similar to the uncertainty weighted models that don’t include gamma either.

7. Simulations: please include simulations in the supplement that show how well individual differences in behaviour can be recovered from data (see e.g. Melinscak and Bach (2019)). Also show to what extend the behavioural measure delta P (figure 7) correlates with the model parameters. As an aside please also explain why this is used instead of the model parameter? I assume it was done because the model was not fitted to each person individually?

8. Data in figure 2: is the average made over the whole task, or only e.g. for the second half, when participants have already learnt? This appears relevant as if the effects get stronger with time (as they appear to do according to figure 4), a clearer picture for the 10% and 90% stimulus might emerge by only using later trials?

FMRI:

9. In figure 8 A-C it appears that there is no representation of the stimulus value in vmPFC. This seems pretty surprising given previous studies. Is it maybe because the regressor used to measure this was fs, rather than e.g. reward probability from a Rescorla-Wagner type model or the participant’s subsequent rating, which should maybe give the clearest prediction of a participant’s current value estimate. If that was the case, as follow-up analyses could be done similar to figure 5 in Palminteri et al. (2015) that use model-comparison to look at the context effects

10. Figure 8 – are only ‘positive effects’/ activations considered? E.g. commonly dACC is found to activate with inverse value at time of decision.

Minor comments:

- Figure 8, it would be helpful if for each contrasts (lacking e.g. for D) it was stated explicitly where this was time locked to (rating or outcome).

- Modeling: one feature of the proposed model I am not quite sure about: for a given trial, the model computes the probability estimate (PS) based on fs , foverall with a weight tau. And in a Rescorla Wagner framework, one would think that this is then stored for future computation. However, it seems that in the framework proposed here, series of outcomes are stored accurately, but then only skewed when one retrieves them to make a decision. If that is the assumption, it would be nice to discuss the rationale for this more. 

- Figure 2, the figure would be easier to read if in A the y-axis was probability

- FMRI: The authors find no difference between the contrasts for the stimuli in the different contexts. One complementary analysis would be analogous to Palminteri et al. where they first identified ROIs based on coding probability (see comments above) and then test in these with model-comparison whether probabilities derived from either model better explain the neural activity

References

Ahn W-Y, Krawitz A, Kim W, Busemeyer JR, Brown JW (2013) A model-based fMRI analysis with hierarchical Bayesian parameter estimation.

Carpenter B, Gelman A, Hoffman MD, Lee D, Goodrich B, Betancourt M, Brubaker M, Guo J, Li P, Riddell A (2017) Stan: A Probabilistic Programming Language. J Stat Softw 76 Available at: https://www.osti.gov/pages/biblio/1430202-stan-probabilistic-programming-language [Accessed July 29, 2019].

Khaw MW, Glimcher PW, Louie K (2017) Normalized value coding explains dynamic adaptation in the human valuation process. Proc Natl Acad Sci 114:12696–12701.

Koehler JJ (1996) The base rate fallacy reconsidered: Descriptive, normative, and methodological challenges. Behav Brain Sci 19:1–17.

Melinscak F, Bach D (2019) Computational optimization of associative learning experiments. Available at: https://osf.io/cgpmh/ [Accessed August 16, 2019].

Palminteri S, Khamassi M, Joffily M, Coricelli G (2015) Contextual modulation of value signals in reward and punishment learning. Nat Commun 6:8096.

Reviewer #2: This is a very interesting paper using mathematical modeling combined with multivariate fMRI to study context effects in probability estimation. As the authors note, there is a growing, but controversial, literature on context effects in valuation, but this paper tackles the other half of the decision-making problem by focusing on probabilities. The authors use a simple but elegant design to present participants with three probabilities (10%, 50%, and 90%) in two different contexts (paired with the other two probabilities). They demonstrate that these pairings produce repulsion effects, in particular for 50%, where 50% is judged as being higher probability when it is paired with 10% vs. 90%. Importantly, the authors go on to show that this distortion in probability estimates carries over to a later binary choice task, biasing participants incentivized choices. As an added bonus, the authors replicate these effects in a second replication experiment. The behavioral effects are striking and an important finding on their own. The authors go on to argue that these effects are best explained by a simple reference-point model, and that they cannot be explained by normalization models from the context-effects-in-valuation literature. Finally, the authors show that the dACC correlates with these behavioral distortions, both at stimulus presentation and at feedback, while the vmpfc does so only at feedback. Also, the temporal cortex tracks the stimulus probability, while the ventral striatum and fusiform track the overall probability within a context; these variables appear to be important for generating the overall probability estimations generated by the participants. 

I think that these are important, interesting, and novel findings. To rigorously test the authors’ proposed model, I would have liked to have seen an experimental design with more than 3 probabilities and 2 levels of uncertainty, but I think that this can be left to future work. I don’t see the precise formulation or validation of the model as a critical element of the paper. The paper puts forth a sensible model and shows that it outperforms other existing (or related) models. If the authors can clarify a few things about their analyses and results, I would be happy to endorse publication. 

Introduction:

Page 6: I don’t really understand this “second hypothesis” the way it is written. Please revise. In particular, what does this mean: “potential context effects in probability estimation that are unique to valuation and thus points to neural systems dissociable from valuation.”? 

Why is there no mention of striatum in the introduction, especially since it plays an important role in the results? 

What does tracking recent and distant reward history have to do with probability, in terms of justifying looking at the dACC?

Results:

For 10% and 90% what are the fractions of participants who show the expected effect (as you show with 50%)? It looks like there might be a couple of substantial outliers here.

For Figure 3, t-tests seem inappropriate since these choice probabilities are not normally distributed. I might suggest a non-parametric test like Mann-Whitney. (Essential)

It seems important to show that participants’ behavior in the fMRI task correlates with their choice bias in the post-fMRI task. (Essential)

In Fig. 4B it looks like the 50% paired with 10% was in fact better early on during pre-fMRI than the one paired with 90%. Could the authors sub-sample their participants in such a way that this is not the case, and then still show the behavioral effect? The control experiment helps, but it would be nice to unambiguously see the effect in both experiments. 

What did you expect to see in the response times? Without a clear hypothesis, these results seem better suited to the supplements.

Control experiment: Why so few participants? Given that the effects at 90% and 10% were previously borderline, I would have expected a slightly larger experiment here, especially given that it is purely behavioral. Also, where are the statistics for the probability estimates? (Essential)

Doesn’t the model make an even stronger prediction, which is that the coefficient on f_s should be the coefficient on -f_overall, plus one? (This actually appears to be roughly true in Fig. 6B). Can the authors formally test this? And shouldn’t the authors be testing that tau is indeed close to sigma? Does the model fit improve with a flexible tau? 

Fig. 6C caption - there is a repeat here of UncRef-var-wp. Also, I would suggest trying to come up with better names or acronyms for these models, rather than these awkward labels.

I’m not sure that I would bother including the normalization fits in the main text. Without more background, these won’t make sense to most readers. Perhaps it could just be a discussion point instead and you could point to results in the supplements? I leave this up to the authors though.

Figure 7 - something needs to be noted here regarding “voodoo correlations”. While these correlations are significant, their magnitude is certainly inflated due to the fact you are looking at the peak correlations from many many tests. One potential way to get around this issue, would be to identify the best voxels using one measure (delta_p) and then only use those voxels to rerun the analysis using the other behavioral measure (i.e. the choice bias in the post-fMRI task). Or vice-versa. (Essential)

On p.23 you repeat the same results twice. You describe the correlations with f_s and f_overall, then do so again. 

The results on reward prediction errors seem to come out of nowhere. It would be useful to briefly explain why you did these analyses. I assume that it was to show that you had enough power to detect probability effects, if they were there? 

Did you test if reward prediction errors were better explained with vs. without the observed context effects?

Discussion:

If f_s is really represented in temporal cortex activity, why don’t participants just report that? Why combine it with f_overall? In other words, if there is an unbiased measure of the stimulus probability, why aren't participants reporting it?

Methods:

What does it mean that “the order of [the visual stimuli] presentation was randomized for each block separately”? Were the blocks randomly ordered or not? It seems they were in the fMRI session, but what about pre-fMRI? (Essential)

Was there any incentivization for the guessing task?

In Model 1 why interact PE with MAG? How did you define PE? 

Is there a citation for this particular type of permutation test, or did you come up with it yourselves? (Essential)

It seems arbitrary to use the last 10 trials to model f and sigma. Why this measure and not the true values, the observed values, or some RL-like version?

Reviewer #3: This interesting manuscript describes an fMRI study testing whether probability estimations and related choices are affected by context, in comparable ways as valuation processes are. In 2 behavioral experiments, the authors show that participants’ probability estimates for a given series of cue-related reward outcomes were indeed systematically biased by the overall probability of reward (manipulated by differing reward probabilities for a second cue presented in the same experimental block). This bias was strongest for intermediate probabilities (50%) and appeared well captured by a new computational model that the authors systematically test against other competing models derived from the valuation literature. Analysis of the fMRI data acquired during one of the experiments showed that the context effects were reflected in multi-voxel activity patterns in the ACC and vmPFC, and that the strength of these neural context effects correlated across participants with the strength of the behavioral context effects. The authors conclude that the vmPFC may constitute a shared neural substrate underlying both probability estimation and valuation and that context may affect both these processes. 

I enjoyed reading this manuscript: It addresses an interesting and relevant question with an innovative experimental approach; the research methods are sophisticated and generally applied in a sound way; the results are reasonably clear-cut; and the manuscript is well-written and provides interesting discussion. However, there are several issues with the motivation, experimental design, procedures, and results that the authors should address to render their manuscript more conclusive and balanced.

(1) The authors focus in their ROI analyses on the OFC/vMPFC and ACC, based on the previous literature. However, it is not clear to the reader whether these are really the only two areas that could be expected to be involved in probability coding, based on a truly a-priori review of the literature on probability coding. Off the top of my head, I would have expected also the anterior insula and the IPS to be consistently found in such studies, but there may be even more areas. To prevent the impression of “post-hoc” hypotheses, and to counteract possible bias, the authors should go over pertinent meta-analyses and test all the ROIs suggested by the prior literature to be involved in probability coding. Reporting these analyses is valuable even if they yield null results. 

(2) The authors find that context effects are strongest for 50% probability, compared to 10% and 90%. The formal model suggests that this differential effect reflects different levels of outcome uncertainty. However, a convincing test of this quantitative hypothesis would require many more probabilities than just the somewhat special cases of 50% and two values at the very extreme ends of the scale (10% and 90%): We know that the latter two probabilities are perceived in distorted ways; moreover, there are ceiling and floor effects that may affect estimations. I do like the new model proposed by the authors and I appreciate the careful model comparisons, but the dataset is not rich enough to fully test the quantitative predictions afforded by the model. Ideally, the authors would conduct a control experiment that also includes intermediate probabilities (say 30% and 70%) for which no strong perceptual distortions have been reported but that nevertheless differ in outcome uncertainty from the special case of 50%. At the very least, the authors should discuss this caveat to not give the impression that all predictions of their interesting new model have been tested by the somewhat restricted experimental design. 

(3) In the fMRI experiment, the authors chose to have a category boundary at 50%, meaning that a perceived probability of 50% forces subjects to choose whether to opt for the higher or the lower response category. The authors show in a behavioral control experiment that this probably did not cause their behavioral effects, but the neuroimaging data may still be affected by it. This is particularly worrisome as the ACC is known to be involved in situations with response conflicts such as those present in the scenario described above (one probability being associated with two competing responses). The authors should discuss this caveat, perhaps focusing on why specifics of the activity patterns they observe (anatomical location, involvement in different stages of the design) support the view that response conflict is less likely to explain their results than their alternative proposal.

(4) The authors constructed different sequences of reward events by random draws from distributions with the desired mean probability. This strategy is good in controlling for many things; however, it invariably means that different participants saw different sequences and were exposed to actual frequencies that differed from the desired target frequency. The authors account for this property of their design in the model they build and in some of their analyses, but not in all of them. For instance, for the analyses presented in Figure 2, were the individual probability estimates correlated with the actual frequencies experienced by each individual? Likewise, how were the effects in presented in Figure 3 correlated with individual frequencies? The authors should present these analyses to convince the reader that their model accurately explains each individual’s frequency perceptions, not just the average tendency in the pooled data.

(5) The authors state that the ventral striatum (VS) codes outcome magnitude and refer to Table 2, but this table does not contain the VS?

(6) The authors fit the model with a moving window extending over the last 10 trials. How did they select this number? Are their results robust to different window lengths?

(7) On page 9, the authors write: “We found significant context effect on probability estimate when reward probability was 50%, but not at 10% (t=1.9419, df=33, p=0.0607) or 90% (t=1.9441, df=33, p=0.0605).“ Are these statistics for the analyses of the data acquired for these two probabilities? In that case, I do not think that p-values of p=0.06 provide convincing evidence that there is no context effect. The authors should show that the effects are significantly stronger for 50% than for the two other probabilities, in direct statistical comparisons of these effects.

(8) The authors present the context effects on neural probability representations (in dACC and vmPFC) and the neural representation of contex (in VS and temporal areas) in isolation, even though these representations should be linked if their model and interpretation is correct. Please report whether the strength of these two sets of effects indeed correlates across subjects as would be expected. The authors may also consider conducting connectivity analyses to analyse how the information is shared between these two sets of regions (but such new analyses are clearly optional and not required for their results to stand).

(9) The manuscript is unclear as to whether the authors used cluster- or voxel-level inference. I suspect the former given that they mention a cluster-forming threshold, but this should be stated explicitly. Moreover, the authors should ensure they properly account for the problems associated with some types of cluster-level inference as highlighted in Eklund et al 2016 PNAS.

(10) On Page 33, the authors write: “It reflects that seeing a stimulus with intermediate probability of reward like 0.5 feels better when the other stimulus present in the context is much worse than 0.5 than when the other stimulus carries a much better chance of reward.” What is the evidence for these presumed feelings? The authors either need to present this or refrain from inferring on feelings based on fMRI data alone.

---

## [Editor Report · Decision Letter 2]

7 Jan 2020

Dear Dr Wu,

Thank you for submitting your revised Research Article entitled "Context effect on probability estimation" for publication in PLOS Biology. I have now discussed your revision with the Academic Editor, and I'm delighted to let you know that we're now editorially satisfied with your manuscript. 

However before we can formally accept your paper and consider it "in press", we also need to ensure that your article conforms to our guidelines. A member of our team will be in touch shortly with a set of requests. As we can't proceed until these requirements are met, your swift response will help prevent delays to publication. Please also make sure to address the data and other policy-related requests noted at the end of this email.

*Copyediting*

*Published Peer Review History*

*Early Version*

*Submitting Your Revision*

Sincerely,

Gabriel Gasque, Ph.D., 

Senior Editor

PLOS Biology

ETHICS STATEMENT:

The Ethics Statements in the submission form and Methods section of your manuscript should match verbatim. Please ensure that any changes are made to both versions.

-- Please indicate if your protocols approved by the Taipei Veteran General Hospital IRB were conducted according to the principles expressed in the Declaration of Helsinki or any other national or international regulations/guidelines.

DATA POLICY:

-- Please also ensure that the figure legends in your manuscript include information on where the underlying data can be found, and ensure your supplemental data file/s has a legend.

-- Please annotate your data and code files in https://osf.io/48j7m sufficiently, so they can be directly linked to each of the figures displaying quantitative data. You can do this by providing a detailed ReadMe file 

--Please ensure that your Data Statement in the submission system accurately describes where your data can be found.

---

## [Editor Report · Decision Letter 3]

14 Feb 2020

Dear Dr Wu,

On behalf of my colleagues and the Academic Editor, Matthew Rushworth, I am pleased to inform you that we will be delighted to publish your Research Article in PLOS Biology. 

Early Version

PRESS 

Kind regards,

Alice Musson

Publication Assistant, 

PLOS Biology

on behalf of

Gabriel Gasque,

Senior Editor

PLOS Biology